# Age-dependent deterioration of nuclear pore assembly in mitotic cells decreases transport dynamics

Irina L Rempel[1], Matthew M Crane[2], David J Thaller[3], Ankur Mishra[4], Daniel PM Jansen[1], Georges Janssens[1], Petra Popken[1], Arman Akşit[1], Matt Kaeberlein[2], Erik van der Giessen[4], Anton Steen[1], Patrick R Onck[4], C Patrick Lusk[3], Liesbeth M Veenhoff[1]*

[1]European Research Institute for the Biology of Ageing (ERIBA), University of Groningen, University Medical Center Groningen, Groningen, Netherlands; [2]Department of Pathology, University of Washington, Seattle, United States; [3]Department of Cell Biology, Yale School of Medicine, New Haven, United States; [4]Zernike Institute for Advanced Materials, University of Groningen, Groningen, Netherlands

**Abstract** Nuclear transport is facilitated by the Nuclear Pore Complex (NPC) and is essential for life in eukaryotes. The NPC is a long-lived and exceptionally large structure. We asked whether NPC quality control is compromised in aging mitotic cells. Our images of single yeast cells during aging, show that the abundance of several NPC components and NPC assembly factors decreases. Additionally, the single-cell life histories reveal that cells that better maintain those components are longer lived. The presence of herniations at the nuclear envelope of aged cells suggests that misassembled NPCs are accumulated in aged cells. Aged cells show decreased dynamics of transcription factor shuttling and increased nuclear compartmentalization. These functional changes are likely caused by the presence of misassembled NPCs, as we find that two NPC assembly mutants show similar transport phenotypes as aged cells. We conclude that NPC interphase assembly is a major challenge for aging mitotic cells.
DOI: https://doi.org/10.7554/eLife.48186.001

*For correspondence:
l.m.veenhoff@rug.nl

## Introduction

Rapid and controlled transport and communication between the nucleus and cytosol are essential for life in eukaryotes and malfunction is linked to cancer and neurodegeneration (reviewed in *Fichtman and Harel, 2014*). Nucleocytoplasmic transport is exclusively performed by the Nuclear Pore Complex (NPC) and several nuclear transport receptors (NTRs or karyopherins) (reviewed in *Fiserova and Goldberg, 2010*; *Hurt and Beck, 2015*). NPCs are large (~52 MDa in yeast and ~120 MDa in humans) and dynamic structures (*Alber et al., 2007*; *Kim et al., 2018*; *Onischenko et al., 2017*; *Teimer et al., 2017*). Each NPC is composed of ~30 different proteins, called nucleoporins or Nups (*Figure 1a*). The components of the symmetric core scaffold are long lived both in dividing yeast cells and in postmitotic cells, while several FG-Nups are turned over (*D'Angelo et al., 2009*; *Denoth-Lippuner et al., 2014*; *Savas et al., 2012*; *Thayer et al., 2014*; *Toyama et al., 2013*) and dynamically associate with the NPC (*Dilworth et al., 2001*; *Niño et al., 2016*; *Rabut et al., 2004*). Previous studies performed in postmitotic aging cells (chronological aging) showed changes in NPC structure and function (*D'Angelo et al., 2009*; *Toyama et al., 2019*), and also in aging mitotic cells (replicative aging) changes in NPCs have been described (*Denoth-Lippuner et al., 2014*;

**Figure 1.** The cellular abundance of some NPC components changes in replicative aging. (a) Cartoon representation of the NPC illustrates different structural regions of the NPC, all FG-Nups are shown in green independently of their localization, the membrane rings in light brown, the inner rings in purple, the outer rings in brown, the mRNA export complex in pink, and the nuclear basket structure in light blue. Adapted with permission from Springer Nature Customer Service Centre GmbH: Springer Nature, Nature, Integrative structure and functional anatomy of a nuclear pore complex, *Kim et al. (2018)*. (b) Schematic presentation of replicative aging yeast cells. (c) Transcript and protein abundance of NPC components (color coded as in *Figure 1a*) as measured in whole cell extracts of yeast cells of increasing replicative age; after 68 hr of cultivation the average replicative age of the cells is 24. Cells were aged under controlled and constant conditions (*Janssens et al., 2015*). See also *Figure 1—figure supplement 1a*. (d) Young cells are trapped in the microfluidic device and bright field images are taken every 20 min to define the cells age and fluorescent images are taken once every 15 hr to detect the protein localization and abundance. Representative images of cells expressing indicated fluorescent protein fusions imaged at the start of the experiment and after 30 hr; their replicative age is indicated. Scale bar represents 5 μm. (e) Heat map representation of the changes in the levels of the indicated GFP- and mCh-tagged Nups at the NE in each yeast cell at increasing age. Each line represents a single cell's life history showing the change in the ratio of the fluorescence from the GFP-tagged Nup over the fluorescence from the mCh-tagged Nup and normalized

*Figure 1 continued on next page*

*Figure 1 continued*

to their ratio at time zero. Measurement of the fluorescence ratios are marked with 'x'; in between two measurements the data was linearly interpolated. The fold changes are color coded on a log 2 scale from −1 to + 1; blue colors indicate decreasing levels of the GFP-fusion relative to mCh. Number of cells in the heatmaps are Nup116-GFP/Nup49-mCh = 67, Nup133-GFP/Nup49-mCh = 94 and Nup100-GFP/Nup49-mCh = 126.

DOI: https://doi.org/10.7554/eLife.48186.002

The following figure supplements are available for figure 1:

**Figure supplement 1.** Cellular protein and mRNA abundance of Nups, NTRs and assembly factors in replicative aging.

DOI: https://doi.org/10.7554/eLife.48186.003

**Figure supplement 2.** The abundance and localization of NPC components in replicative aging.

DOI: https://doi.org/10.7554/eLife.48186.004

**Figure supplement 3.** Models of NPCs with altered stoichiometry.

DOI: https://doi.org/10.7554/eLife.48186.005

*Lord et al., 2015*). To study the fate of NPCs in mitotic aging, we use replicative aging budding yeast cells as a model. Individual yeast cells have a finite lifespan which is defined as the number of divisions that they can go through before they die: their replicative lifespan (reviewed in *Longo et al., 2012*) (*Figure 1b*). The divisions are asymmetric and while the mother cell ages the daughter cell is born young. Remarkably, studying the lifespan of this single-cell eukaryote has been paramount for our understanding of aging (reviewed in *Denoth Lippuner et al., 2014*; *Longo et al., 2012*; *Nyström and Liu, 2014*) and many of the changes that characterize aging in yeast are shared with humans (*Janssens and Veenhoff, 2016b*). In the current study, we address changes to the NPC structure and function during mitotic aging by imaging of single cells.

## Results

### The cellular abundance of specific NPC components changes in replicative aging

We previously generated the first comprehensive dynamic proteome and transcriptome map during the replicative lifespan of yeast (*Janssens et al., 2015*), and identified the NPC as one of the complexes of which the stoichiometry of its components changes strongly with aging. Indeed, the proteome and transcriptome data give a comprehensive image of the cellular abundance of NPC components in aging (*Figure 1c*). We observe that the cellular levels of NPC components showed loss of stoichiometry during replicative aging, which were not reflected in the more stable transcriptome data (*Figure 1c*; *Figure 1—figure supplement 1a*). Clearly in mitotic aging, a posttranscriptional drift of Nup levels is apparent.

The total abundance of NPC components measured in these whole cell extracts potentially reflects an average of proteins originating from functional NPCs, prepores, misassembled NPCs, and possibly protein aggregates. Therefore, we validated for a subset of Nups (Nup133, Nup49, Nup100, Nup116 and Nup2) that GFP-tagged proteins expressed from their native promoters still localized at the nuclear envelope in old cells. In addition, we validated that changes in relative abundance of the Nups at the nuclear envelope were in line with the changes found in the proteome. We included Nup116 and Nup2 in our experiments as those Nups showed the strongest decrease in abundance (*Figure 1c*). Nup133 was included because its abundance was stable in aging and Nup100 was included because it is important for the permeability barrier (*Lord et al., 2015*; *Popken et al., 2015*). We used Nup49-mCh as a reference in all of our microfluidic experiments as Nup49 had previously been used as a marker for NPCs. The proteome data indicated that Nup49 showed a relatively stable abundance profile in aging (*Figure 1—figure supplement 1d*). The tagging of the Nups with GFP and mCherry (mCh) reduced the fitness of those strains to different extents but all retained median division time under 2.5 hr (*Figure 1—figure supplement 2b*). Nsp1 could not be included in the validation, because the Nsp1-GFP fusion had a growth defect and could not be combined with Nup49-mCh, Nup100-mCh or Nup133-mCh in the BY4741 background. We used microfluidic platforms that allow uninterrupted life-long imaging of cells under perfectly controlled constant conditions (*Crane et al., 2014*) (*Figure 1d*). The single-cell data of cells expressing

GFP-fusions of Nup133, Nup100 and Nup116 together with Nup49-mCh are shown in *Figure 1e* (see *Figure 1—figure supplement 2c–e* for Nup2 and a tag-swap control). Consistent with the proteome data, and with previously reported data (*Lord et al., 2015*), in the vast majority of aging cells the abundance of Nup116-GFP decreased relative to Nup49-mCh, while the abundance of Nup133-GFP appears more stable. Also for the other Nups tested (Nup100 and Nup2), the imaging data align well with the proteome data (*Figure 1—figure supplement 2d*).

Our data contain full life histories of individual cells and, in line with previous reports (*Crane et al., 2014*; *Fehrmann et al., 2013*; *Janssens and Veenhoff, 2016a*; *Jo et al., 2015*; *Lee et al., 2012*; *Zhang et al., 2012*), we observed a significant cell-to-cell variation in the lifespan of individual cells, as well as variability in the levels of fluorescent-tagged proteins. Therefore, we could assess if the changes observed for the individual NPC components correlated to the lifespan of a cell and, indeed, for Nup116 and Nup100 such correlations to lifespan were found, where those cells with lowest levels of NE-localized GFP-tagged Nups had the shortest remaining lifespan (for Nup100 r $= -0.48$; p$=1.27{\times}10^{-7}$ and Nup116 r $= -0.56$; p$=6.54{\times}10^{-4}$, see *Figure 1—figure supplement 2f,g*). The statistics of these correlations are in line with aging being a multifactorial process where the predictive power of individual features is limited. In comparison to the aging related increase in cell size (a Pearson correlation of around 0.2) (*Janssens and Veenhoff, 2016a*), the correlations found here are relatively large.

Taken together, we confirmed the loss of specific FG-Nups by quantifying the localization and abundance of fluorescently-tagged Nups in individual cells during their entire lifespan. Single-cell Nup abundances at the NE can be highly variable (Nup2), while for other Nups (Nup100, Nup116) the loss in abundance at the NE was found in almost all aging cells and correlated with the lifespan of the cell. From the joint experiments published by *Janssens et al. (2015)*; *Lord et al. (2015)* and the current study we can conclude that especially Nup116 and Nsp1 (Nup98 and Nup62 in humans) strongly decrease in aging.

## Mitotic aging is associated with problems in NPC assembly rather than oxidative damage

A possible cause for the loss of stoichiometry could be that NPCs are not well maintained in aging. Indeed, in postmitotic cells, oxidative damage was proposed to lead to the appearance of carbonyl groups on Nups inducing more permeable NPCs (*D'Angelo et al., 2009*). We have limited information on the maintenance of existing NPCs during replicative aging but there is some precedent for the hypothesis that even in the fast dividing yeast cells damage to existing NPCs may accumulate in aged cells. Indeed, NPCs remain intact during multiple divisions (*Colombi et al., 2013*; *Denoth-Lippuner et al., 2014*; *Khmelinskii et al., 2012*; *Thayer et al., 2014*), and especially in aged mother cells a fraction of the NPCs is inherited asymmetrically to the aging mother cell (*Denoth-Lippuner et al., 2014*; *Shcheprova et al., 2008*). Oxidative stress and reactive oxygen species (ROS) production in the cell is a major source of damage and can result in irreversible carbonylation of proteins (*Stadtman and Levine, 2003*). Protein carbonyls can be formed through several pathways. Here, we focused on the most prominent one, the direct oxidation of the Lysine, Threonine, Arginine and Proline (K, T, R, P) side chains through Metal Catalyzed Oxidation (MCO) (*Stadtman and Levine, 2003*) by the Fenton reaction (*Maisonneuve et al., 2009*; *Stadtman and Levine, 2003*). Despite extensive efforts and using different in vitro and in vivo oxidative conditions and using different carbonyl-detection methods we could not find evidence for oxidative damage of Nsp1, Nup2, Nic96 and Nup133 (*Figure 2—figure supplement 1a,b* shows negative results for Nsp1 along with a positive control).

Further indication that oxidative damage is unlikely to impact the NPC in aging came from modeling studies. We carried out coarse-grained molecular dynamics simulations using our previously developed one-bead-per-amino-acid model of the disordered phase of the NPC (*Ghavami et al., 2013*; *Ghavami et al., 2014*). Earlier studies have shown that this model faithfully predicts the Stokes radii for a range of FG-domains/segments (*Ghavami et al., 2014*; *Yamada et al., 2010*), as well as the NPC's size-dependent permeability barrier (*Popken et al., 2015*). To model the carbonylated FG-Nups, we incorporated the change in hydrophobicity and charge for carbonylated amino-acids (T, K, R, P) into the coarse-grained force fields (see Materials and methods) and modeled maximally carbonyl-modified FG-Nups and NPCs. Overall, there is a minor impact of carbonylation on the predicted Stokes radius of the individual Nups and

the time-averaged density of a wild type and fully oxidized NPC, with average densities around 80 mg/ml and maximum densities reaching 100 mg/ml in the center of the NPC ($r < 5$ nm) (*Figure 2a* red line, *Figure 2b* right panel and see *Figure 2—figure supplement 1c–e* for individual Nups and additional models dissecting the relative impact of the change in charge and hydrophobicity upon carbonylation).

Altogether, we find no experimental evidence for carbonyl modification of FG-Nups even under strong oxidative conditions and, based on our modeling studies, the carbonylation of FG-Nups is predicted to have little, or no impact on the passive permeability of NPCs, even under the unrealistic condition that all FG-Nups are fully carbonylated. We conclude that oxidative damage is unlikely to be a direct cause of altered NPC stoichiometry in replicative aging, and it is probable that the previously reported increase in permeability of NEs during chronological aging (*D'Angelo et al., 2009*) is actually caused by factors other than carbonylation.

We then addressed, if a main driver of NPC decline in replicative aging may be caused by the inability to control de novo NPC assembly. In young and healthy yeast cells, phenotypes associated with misassembled NPCs are rarely seen, but mutant strains with impaired NPC assembly show that a fraction of their NPCs cluster, are covered by membranes, or cause herniations of the NE (*Chadrin et al., 2010*; *Scarcelli et al., 2007*; *Webster et al., 2014*; *Webster et al., 2016*; *Zhang et al., 2018*) (reviewed in *Thaller and Patrick Lusk, 2018*). Misassembled NPCs that are induced by mutations are asymmetrically retained, and accumulated in the mother cell over time (*Colombi et al., 2013*; *Makio et al., 2013*; *Webster et al., 2014*). We thus asked, if replicatively aged cells start to progressively accumulate misassembled NPCs. Correct NPC assembly is assisted by several proteins that are temporarily associated with NPCs during the assembly process (*Dawson et al., 2009*; *Lone et al., 2015*; *Otsuka and Ellenberg, 2018*; *Scarcelli et al., 2007*; *Webster et al., 2016*; *Zhang et al., 2018*). Amongst these are (i) Heh1 and Heh2, the orthologues of human LEM2 and Man1, which have been proposed to recognize misassembled pores (*Thaller et al., 2019*; *Webster et al., 2014*; *Webster et al., 2016*), (ii) Vps4, an AAA-ATPase with multiple functions amongst which the clearance of misassembled NPCs from the NE (*Webster et al., 2014*) and (iii) Apq12, Rtn1 and Rtn2, Brr6 and Brl1 membrane proteins of the NE-ER network that are involved in NPC assembly, possibly through roles in modulating membrane curvature (*Lone et al., 2015*; *Scarcelli et al., 2007*; *Zhang et al., 2018*).

The system wide proteomics data showed that the protein levels of Heh1, Rtn1 and Rtn2 are stable in abundance in aging, while a sharp decrease in abundance was found for Vps4 (*Figure 2c*, and *Figure 1—figure supplement 1c* showing stable transcript levels). Additionally, we found that the abundance of Heh2-GFP, Brl1-GFP and Apq12-GFP at the NE decreased relative to Nup49-mCh in aging (*Figure 2d* and *Figure 1—figure supplement 1c* showing stable transcript levels). Despite the fact that neither Heh2 nor Apq12 are essential proteins, we found their levels to be correlated with the remaining lifespan of the cells, where those cells showing the lowest levels of Heh2-GFP or Apq12-GFP had the shortest remaining lifespan (*Figure 2e* and *Figure 2—figure supplement 2*). The level of the essential protein Brl1 similarly correlated with the remaining lifespan of the cells (*Figure 2f*). Previous work showed that the deletion of either *heh2*, *vps4* or *apq12* is sufficient to cause the appearance of misassembled NPCs in haploid cells (*Scarcelli et al., 2007*; *Webster et al., 2014*) so the decrease in abundance of the proteins Heh2, Apq12, Brl1 and Vps4 suggests that NPC assembly is compromised in aging and misassembled NPCs may accumulate.

To get a more direct readout of problems in NPC assembly. we studied Chm7, the nuclear adaptor for the ESCRT system (*Gu et al., 2017*; *Olmos et al., 2016*; *Webster et al., 2016*). Chm7 sometimes forms a focus at the NE and the frequency of focus formation is related to NPC assembly problems as mutant strains with impaired NPC assembly show more frequently Chm7 foci at the NE (*Webster et al., 2016*). We quantified the frequency of focus formation in differently aged cells. Indeed, the foci are more than twice as frequently seen in the highest age group (age 15–24), compared to cells younger than five divisions. Also, the frequency at which cells have more than one focus present at the NE is more than fourfold higher in the oldest age group (*Figure 2g*). The increased frequency of Chm7 foci in aged cells supports that aged cells have problems in NPC assembly. As misassembled NPC can cause herniations at the NE, which can be observed in EM (*Thaller and Patrick Lusk, 2018*; *Webster et al., 2014*; *Webster et al., 2016*; *Wente and Blobel, 1993*), we quantified the appearance of NE herniations in young and aged cells. In young cells NE

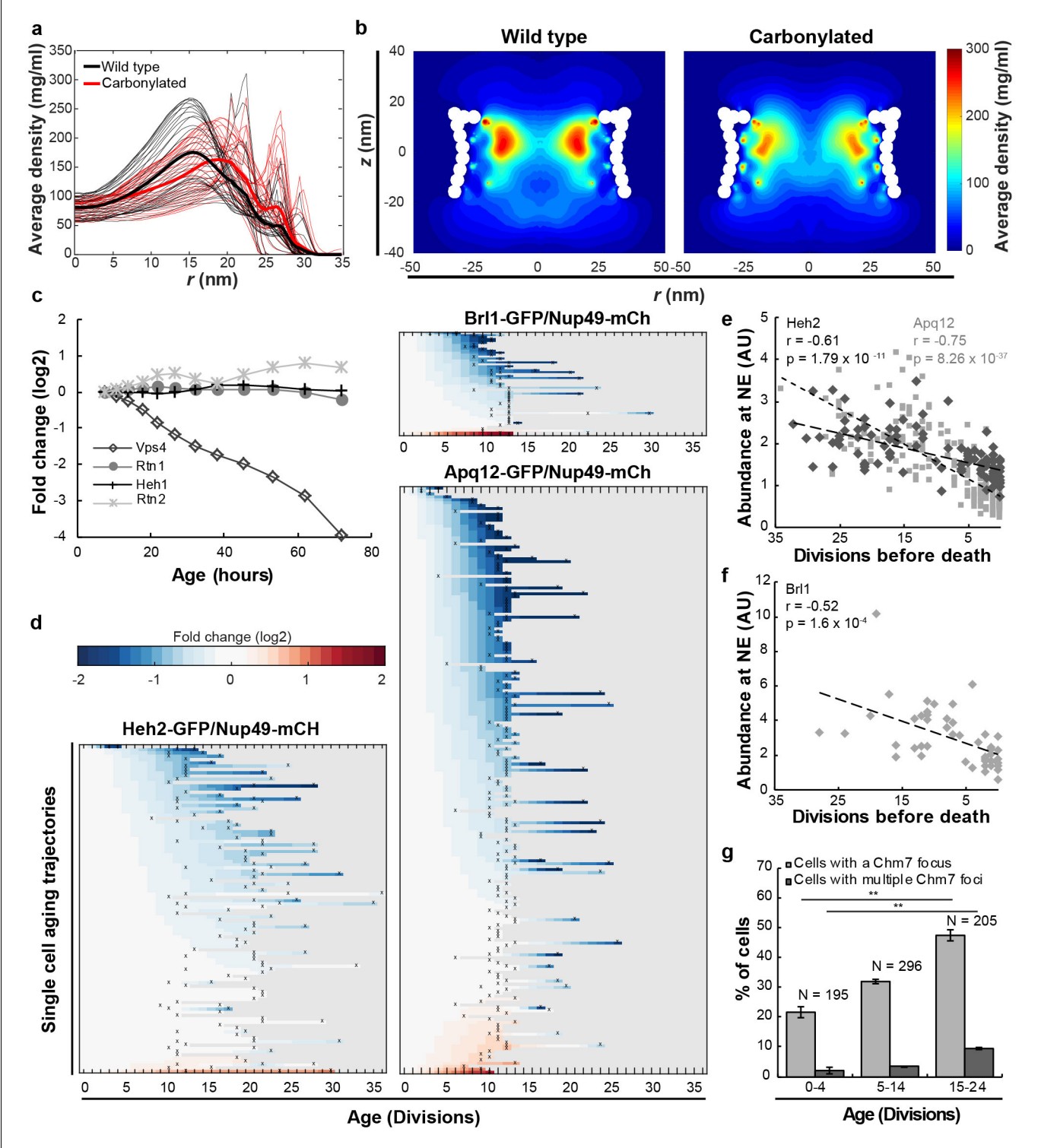

**Figure 2.** Mitotic aging is associated with problems in NPC assembly rather than oxidative damage. (a) Time-averaged radial density distribution of FG-Nups for different positions along the z-axis separated by 1 nm, in the range −15.4 < z < 15.4 nm, plotted for the wild type (black), the maximally carbonylated NPC (red) (See also *Figure 2—figure supplement 1d,e*). The dark colored lines represent the density averaged over the range −15.4 < z < 15.4 nm. (b) Time-averaged r-z density of FG-Nups in the wild type NPC (left panel), the oxidized NPC (right panel). (c) Protein abundance of Heh1, Vps4, Rtn1 and Rtn2 as measured in whole cell extracts of yeast cells of increasing replicative age. Data from *Janssens et al. (2015)*. (d) Heatmaps showing single-cell abundance of Heh2-GFP (N = 100), Brl1-GFP (N = 53) and Apq12 (N = 200) at the NE, relative to Nup49-mCh in replicative aging. (e) Heh2-GFP and Apq12-GFP abundance at the NE, relative to Nup49-mCh, as a function of remaining lifespan. The dotted lines

*Figure 2 continued on next page*

*Figure 2 continued*

indicate best linear fit; Pearson correlations are indicated. Number of cells analyzed are Apq12 = 82, Heh2 = 51 and number of measuring points analyzed are Apq12 = 193 and Heh2 = 102. Data represents single replicates, a second replicate is shown in *Figure 2—figure supplement 2*. (f) Brl1 abundance at the NE, relative to Nup49-mCh, as a function of remaining lifespan. The dotted lines indicate best linear fit; Pearson correlations are indicated. Number of cells analyzed are 20 and number of measuring points analyzed are 47. (g) Percentage of cells with a Chm7 focus reflecting faulty NPCs at the NE at different ages. Buds were excluded from the analysis. Error bars are weighted SD from the mean, from three independent replicates. p-Values from Student's t-test **p≤0.01. N = Total number of cells.

DOI: https://doi.org/10.7554/eLife.48186.006

The following figure supplements are available for figure 2:

**Figure supplement 1.** In vitro oxidation and models of NPCs with oxidative damage.

DOI: https://doi.org/10.7554/eLife.48186.007

**Figure supplement 2.** Heh2-GFP and Apq12-GFP abundance at the NE as a function of remaining lifespan.

DOI: https://doi.org/10.7554/eLife.48186.008

herniations are found in only 2% of the nuclei. In aged cells, those herniations are found much more frequently, with 17% of the nuclei showing a herniation (*Figure 3a,b*).

We conclude that four proteins involved in the assembly of NPCs decrease strongly in abundance in aging (Vps4, Heh2, Brl1 and Apq12) in a manner that correlates with remaining lifespan (*Figure 2*). Jointly, the decrease in abundance of those proteins, and potentially also the decrease of FG-Nup abundance (*Figure 1*), likely directly cause the NPC assembly problems, which we observe as an increased Chm7 focus formation frequency (*Figure 2g*) and an increased number of herniations (*Figure 3*) in aged cells.

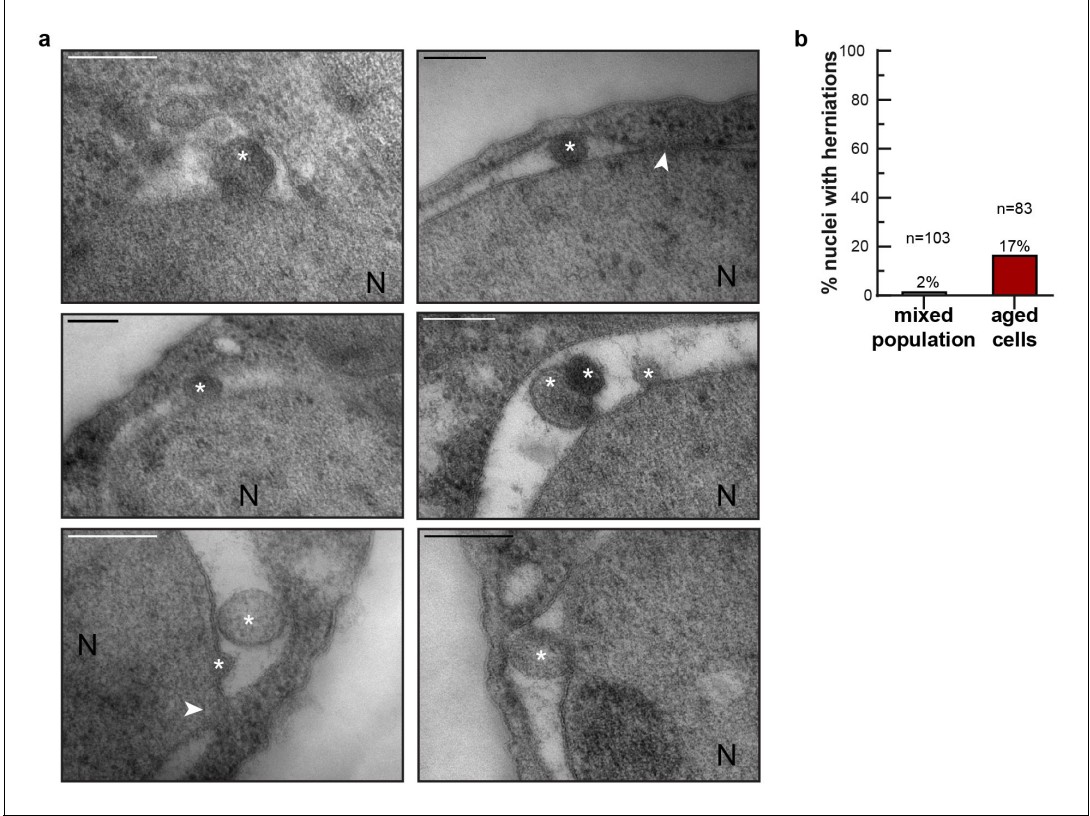

**Figure 3.** NE herniations are more prevalent in aged cells. (a) Examples of NE herniations found in replicatively aged cells. NPCs are indicated by an arrowhead, asterisks indicate herniation lumens and the nucleus is marked with N. Scale bars are 200 nm. (b) Quantification of nuclei with herniations in thin sections. n indicates the number of cells with a visible nucleus analyzed.

DOI: https://doi.org/10.7554/eLife.48186.009

## Increased steady state nuclear compartmentalization in aging is mimicked in NPC assembly mutants

Next, we experimentally addressed the rates of transport into and from the nucleus with aging. During import and export, NTRs bind their cargoes through a nuclear localization signal (NLS) or nuclear export signal (NES) and shuttle them through the NPC. In addition to facilitating active transport, the NPC is a size dependent diffusion barrier (*Popken et al., 2015*; *Timney et al., 2016*). We measured the rate of efflux in single aging cells and find that passive permeability is not altered significantly in aging (*Figure 4—figure supplement 1a–c*), excluding the possibility that NPCs with compromised permeability barriers ('leaky' NPCs) are prevalent in aging cells.

We then looked at classical import facilitated by the importins Kap60 and Kap95, and export facilitated by the exportin Crm1. The cellular abundance of Crm1, Kap60 and Kap95 is relatively stable in aging (*Janssens et al., 2015*) (*Figure 4—figure supplement 2a* and *Figure 1—figure supplement 1c* for transcript levels) as is their abundance at the NE and their localization (*Figure 4—figure supplement 2b–d*). To test whether their transport changes with aging, we used GFP-tcNLS (GFP with a tandem classical NLS, Kap60 and Kap95 import cargo) (*Goldfarb et al., 1986*; *Wychowski et al., 1985*) and GFP-NES (Crm1 export cargo) (*Shulga et al., 1999*) reporter proteins, and GFP as a control. We carefully quantified the steady-state localization of transport reporters in individual aging cells in the non-invasive microfluidic setup (See *Figure 4—figure supplement 3* for lifespan of strains). In the vast majority of cells, we observed that GFP carrying a tcNLS accumulated more strongly in the nucleus at high ages (*Figure 4a*, middle panel), and, interestingly, the GFP carrying a NES is more strongly depleted from the nucleus in the vast majority of cells (*Figure 4a*, right panel). For the control, GFP, we find a more stable N/C ratio in aging (*Figure 4a*, left panel). While the changes in steady state accumulation are observed already early in life when looking at single cells, on the population level the changes become significant only later in the lifespan (*Figure 4b*). To see whether an increase in nuclear compartmentalization in aging was reproducible across different signal sequences, we further quantified the localization of reporter proteins that carried a Nab2NLS (Kap104 import cargo), or a Pho4NLS (Kap121 import cargo) (*Kaffman et al., 1998*; *Timney et al., 2006*; *Truant et al., 1998*). Also for these two signal sequences, we found that reporter proteins with the respective sequences accumulated more strongly in the nucleus at higher ages (*Figure 4c* and *Figure 4d*).

How should we interpret the increased steady state localization of these 4 GFP reporters in aging? The steady state localization of these GFP-reporter proteins depends on the kinetics of NTR facilitated transport (import or export) and passive permeability (influx and efflux). While we cannot formally exclude that retention mechanisms appear during aging, the efflux experiments in *Figure 4—figure supplement 1a–c* do confirm that GFP remains mobile in aged cells, and also the stable localization of the control, GFP (*Figure 4a*), supports that retention mechanisms have little impact. Thus, under the assumption that retention mechanisms play an age-independent and minimal role, we can interpret the steady state ratio's to report on the balance between the rates of NTR-facilitated-transport (import and export) and passive permeability (influx and efflux). This would mean that the systematic changes in the steady state localization of the reporter proteins that we observe in the aging cells results from a change in the balance between the rates of NTR-facilitated-transport and passive permeability.

Changes in the rates of NTR-facilitated-transport and passive permeability may be related to changes in the NPCs themselves or they may be related to an increased availability of NTRs. We measure no changes in abundance of NTRs (*Figure 4—figure supplement 2a*) and find no indication that the abundance of protein cargo changes during aging *Figure 4—figure supplement 2e,f*). Moreover, the increased nuclear compartmentalization seems to be independent of the reporter protein's respective NTRs. We thus consider it less likely that the rates of NTR-facilitated-transport and passive permeability are related to an increased availability of NTRs and further explore how changes in the NPCs can explain the altered balance between the rates of NTR-facilitated-transport (import and export) and passive permeability (influx and efflux).

To our knowledge, mutation or deletion of Nup53 is the only mutation in the NPC that has been shown to lead to increased steady state compartmentalization (of Kap121 dependent cargo) (*Makhnevych et al., 2003*). On the contrary, many strains, including those where NPC components that decrease in abundance in aging are deleted or truncated, show loss of compartmentalization

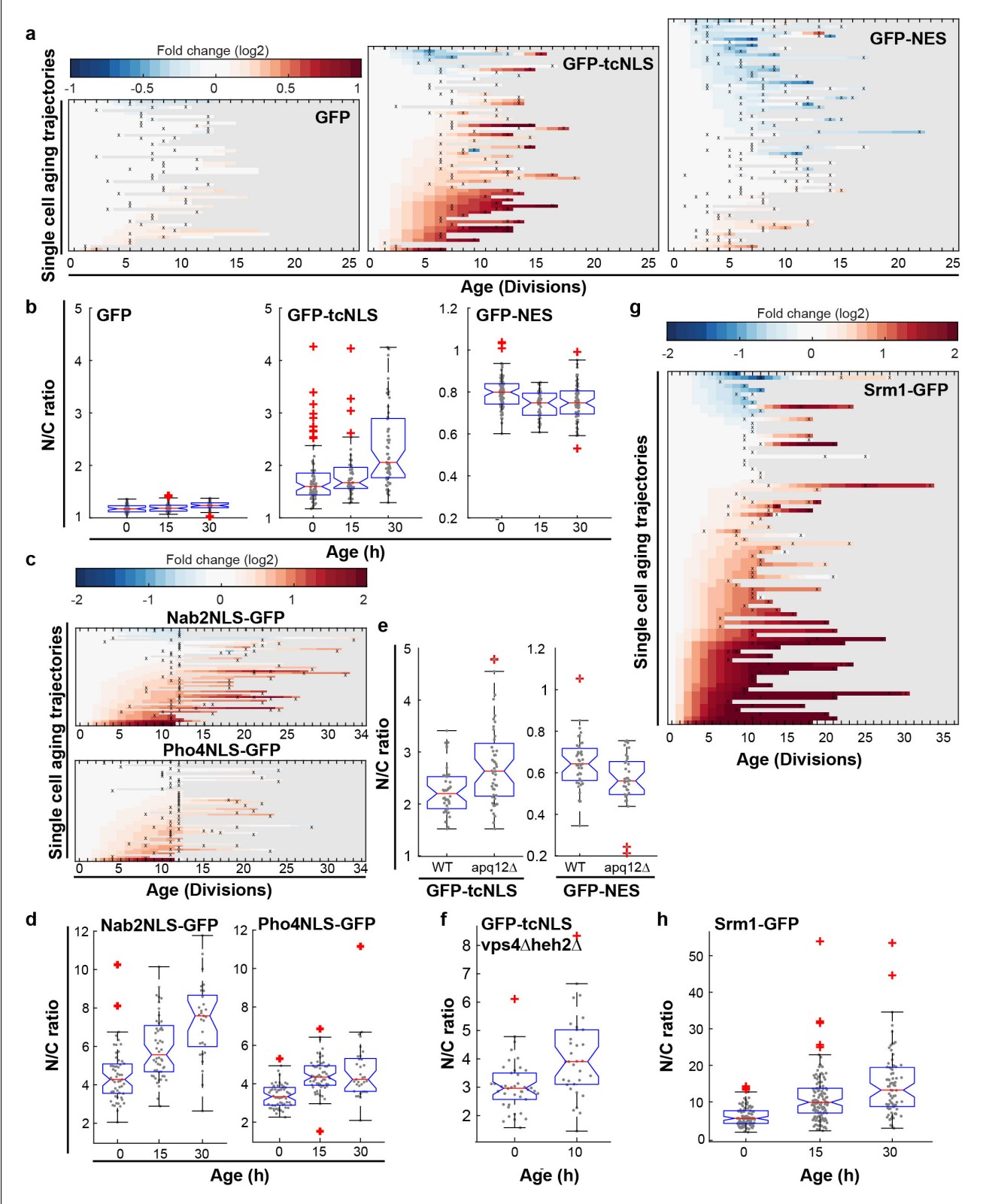

**Figure 4.** Increased steady state nuclear compartmentalization in aging is mimicked in NPC assembly mutants. (a) Heatmaps showing single-cell changes in localization (N/C ratios) of GFP (N = 49), GFP-NES (N = 75) and GFP-NLS (N = 66) reporter proteins during replicative aging. (b) N/C ratios of GFP-tcNLS, GFP-NES and GFP as the cells age. The line indicates the median, and the bottom and top edges of the box indicate the 25th and 75th percentiles, respectively. The whiskers extend to the data points, which are closest to 1.5 times above below the inter quartile range, data points above

*Figure 4 continued on next page*

*Figure 4 continued*

or below this region are plotted individually. Non-overlapping notches indicate that the samples are different with 95% confidence. The number of cells analyzed are GFP = 54, 51, 34; GFP-NLS = 74, 48, 57 and GFP-NES = 75, 41, 66 at time points 0 hr, 15 hr and 30 hr, respectively. (c) Heatmaps showing single-cell changes in localization (N/C ratios) of Nab2NLS-GFP (N = 53) and Pho4NLS-GFP (N = 56) reporter proteins during replicative aging. (d) Median N/C ratios of Nab2NLS-GFP and Pho4NLS-GFP as the cells age. The number of cells analyzed are Nab2NLS-GFP = 55, 52, 29 and Pho4NLS-GFP = 59, 58, 33 at time points 0 hr, 15 hr and 30 hr, respectively. (e) Deletion of *apq12* increases nuclear compartmentalization of GFP-NLS and GFP-NES. The number of cells analyzed are GFP-NLS = 42, 48 and GFP-NES = 39, 34 for WT and Δapq12, respectively (f) Increased nuclear compartmentalization of GFP-NLS during early aging (10 hr of aging, median age of 2 divisions) in a Δvps4Δheh2 background. The number of cells analysed are 42 and 33, respectively. (g) Heatmap showing single-cell changes in localization (N/C ratios) of Srm1-GFP (N = 85) during replicative aging. (h) N/C ratios of Srm1-GFP increases as cells age. Numbers of cells analysed are N = 103, 125, 77 at time points 0 hr, 15 hr and 30 hr, respectively.

DOI: https://doi.org/10.7554/eLife.48186.010

The following figure supplements are available for figure 4:

**Figure supplement 1.** Efflux rate constants in aging.
DOI: https://doi.org/10.7554/eLife.48186.011
**Figure supplement 2.** The abundance of transport factors and NTR cargos does not change in aging.
DOI: https://doi.org/10.7554/eLife.48186.012
**Figure supplement 3.** Replicative lifespan curves.
DOI: https://doi.org/10.7554/eLife.48186.013
**Figure supplement 4.** Apq12 is an essential gene in BY4741, but not in W303.
DOI: https://doi.org/10.7554/eLife.48186.014

(*Lord et al., 2015*; *Popken et al., 2015*; *Strawn et al., 2004*). Interestingly, the only other strain that was previously reported to have an increased compartmentalization is a strain defective in NPC assembly due to a deletion of *apq12* (*Scarcelli et al., 2007*; *Webster et al., 2016*). We found that deletion of *apq12* is genomically instable and not viable in the BY strain background (*Figure 4—figure supplement 4*), hence we recreated the deletion mutant in the W303 background, where it is stable. Indeed, we found that the deletion of *apq12* was sufficient to mimic the increase in compartmentalization seen in aging showing increased nuclear accumulation of GFP-NLS and exclusion of GFP-NES (*Figure 4e*). To further investigate whether the accumulation of misassembled NPCs could cause an increase in nuclear compartmentalization, we quantified the localization of GFP-NLS in a *vps4Δheh2Δ* double mutant. Both individual mutations were previously shown to progressively accumulate misassembled NPCs during aging (*Webster et al., 2014*). We found indeed that cells at a median age of two divisions had a significantly higher N/C ratio of GFP-NLS than young cells (*Figure 3f*). The increased compartmentalization in aged cells and in the *apq12* and *vps4Δheh2Δ* mutant can be explained if fewer functional NPCs are present in the NE. Reduced numbers of NPCs would predominantly impact passive permeability, as the rate-limiting step for NTR-facilitated-transport is not at the level of the number of NPCs but rather at the level of NTRs and cargos finding each other in the crowded cytosol with overwhelming nonspecific competition (*Meinema et al., 2013*; *Riddick and Macara, 2005*; *Smith et al., 2002*; *Timney et al., 2016*).

A previous report showed a reduction in nuclear accumulation of GFP-NLS in age 6 + yeast cells isolated from a culture (*Lord et al., 2015*), while we see no statistically significant difference at this age. We note that there are many differences in the experimental setups that may explain the difference. One possible explanation is that Lord et al., used a different strain background, which might be differently susceptible to NPC assembly problems in aging. Indeed, we noted that several phenotypes, for example the appearance of SINC (Storage of Improperly assembled Nuclear pore complex Compartment; *Webster et al., 2014*) structures related to NPC assembly, are distinct in both strain backgrounds used.

Next, we addressed the transport of native proteins in aged cells. We studied Srm1, the yeast homologue of Rcc1, as endogenously expressed GFP-tagged protein. Srm1 is the nucleotide exchange factor that exchanges GDP for GTP on Ran and its nuclear localization ensures that Ran-GTP levels inside the nucleus are high. The localization of Srm1 depends on Kap60/Kap95-mediated import and retention inside the nucleus via chromatin binding (*Li et al., 2003*; *Nemergut et al., 2001*). While the cellular abundance of Srm1 was stable during aging (*Figure 1—figure supplement 1b*), we found that the nuclear accumulation of Srm1-GFP increased during replicative aging in most cells (*Figure 4g,h*). The steady-state localization of Srm1 cannot be directly interpreted in terms of

transport as retention plays an important role, but it is striking that the change in localization of Srm1-GFP is in line with the changes observed for GFP-NLS. Interestingly, the human homologue of Srm1, Rcc1, was previously also reported to have an increased nuclear concentration in myonuclei and brain nuclei of aged mice (*Cutler et al., 2017*).

## Alterations of the nuclear envelope permeability during aging affects transcription factor dynamics

Additionally, we studied Msn2, a transcriptional regulator that responds to various stresses and translocates to the nucleus in pulses, a so-called frequency modulated transcription factor. Msn2

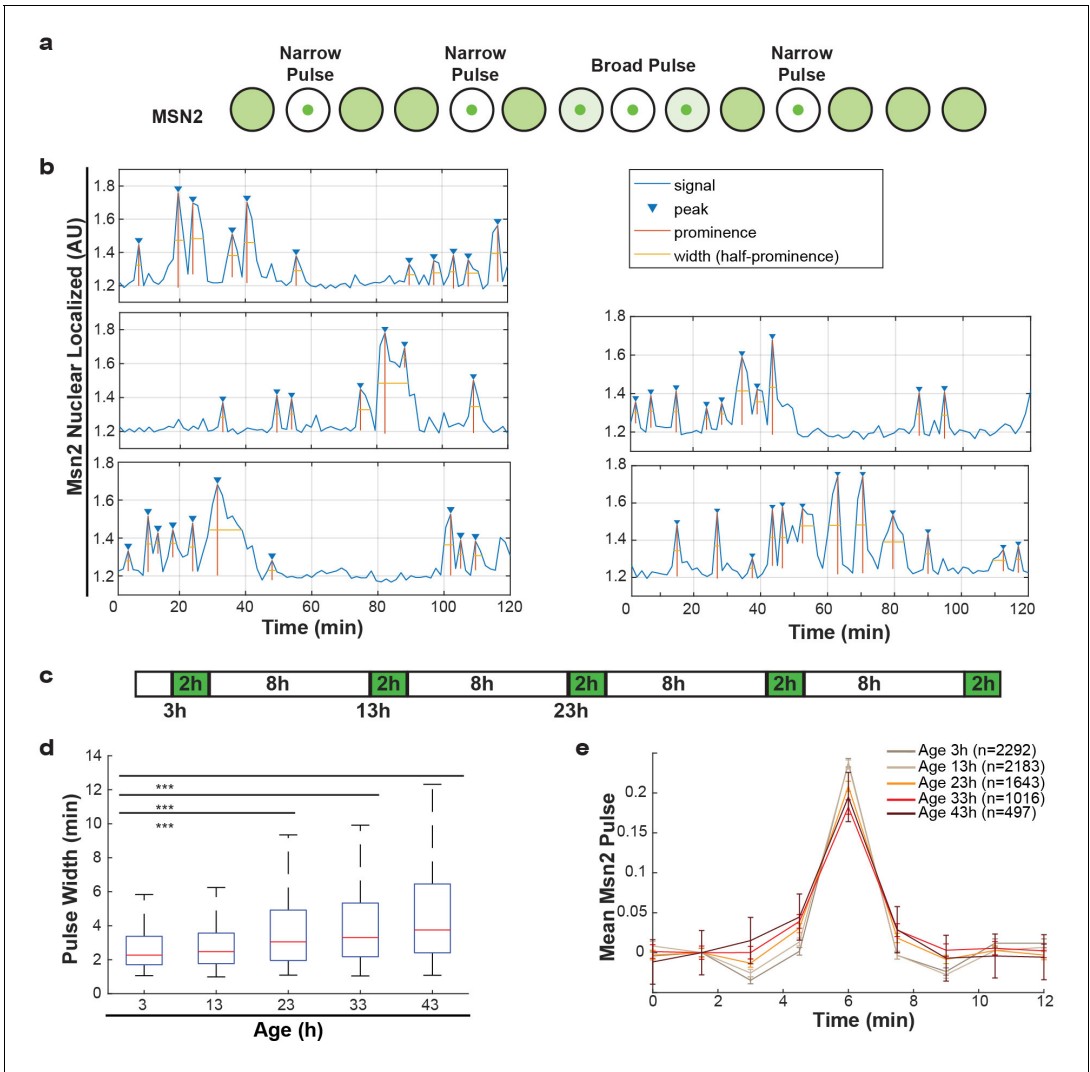

**Figure 5.** Alterations of nuclear envelope permeability during aging affects transcription factor dynamics. (**a**) Schematic showing pulses of Msn2 translocation to the nucleus and movement back to the cytoplasm. (**b**) Five randomly selected single cell traces showing Msn2 dynamics. Low values indicate the majority of Msn2 is cytoplasmic, and high values indicate the majority of Msn2 is nuclear localized. Pulses are annotated showing peaks, prominence and the width of the pulse. (**c**) Experimental protocol for the aging experiment. White boxes indicate brightfield imaging only, and green boxes indicate fluorescence imaging. (**d**) As cells age, the width of the Msn2 pulses increases reliably. (*** indicates p<0.0001 two-tailed t-test). (**e**) Msn2 pulses were identified at each age, and then all pulses were averaged together at each age. To correct for changes in baseline localization with age, the mean pre-pulse level was subtracted at each age. Error bars are standard error.

DOI: https://doi.org/10.7554/eLife.48186.015

The following figure supplement is available for figure 5:

**Figure supplement 1.** Msn2 pulse prominence and width correlate to remaining lifespan.
DOI: https://doi.org/10.7554/eLife.48186.016

communicates information about the environment by modifying the frequency of pulses (*Figure 5a*) (*Cai et al., 2008*; *Hao and O'Shea, 2011*; *Hao et al., 2013*). These pulse dynamics are primarily determined by the rates of nuclear import and nuclear export (*Hao et al., 2013*), but retention mechanisms also apply. By following endogenously tagged Msn2-GFP, we were able to observe the pulse dynamics for individual cells, and quantify specific features of each pulse (*Figure 5b*). Specifically, for each Msn2 pulse, we determined the peak prominence and the pulse width. To determine how aging affected the import and export kinetics, we imaged mother cells for short periods of time every 10 hr (*Figure 5c*). We observed that, as predicted by the alterations in NPCs, the average pulse widths for each cell increased consistently from middle-age onwards (*Figure 5d*). Similarly, by aligning all Msn2 pulses on top of each other for a given age, we determined a mean pulse shape for each age (*Figure 5e*). These show similar characteristics, where the aging results in both broader and lower Msn2 pulses. As could be expected given the striking age-related changes, both the pulse width and pulse prominence are correlated to the remaining lifespan of the cell (*Figure 5—figure supplement 1a,b*, r = −0.36, p<0.0001 and r = 0.47, p<0.0001 for pulse prominence and pulse width, respectively). We consider it likely, that the decrease in Msn2 dynamics is a readout for the NPC's ability to facilitate rapid responses. Aging cells may thus respond more slowly to changes in their environment due to reduced nuclear cytoplasmic transport and communication.

## Discussion

NPC function in aging has received much attention in the context of postmitotic (chronologically) aging cells such as neurons and indeed a large body of data now implicate NPC function in neurodegenerative diseases. Here, we study the fate of NPCs in dividing yeast cells with the anticipation that the insights may be relevant to aging of mitotic cells such as stem cells. Overall, our data is consistent with a model where NPC assembly and quality control are compromised in mitotic aging (*Figure 2*) and where misassembled NPCs, which specifically lack the FG-Nups that we see declining in aging (*Figure 1*), accumulate in aged cells (*Figure 3*). Without intervention, the loss of FG-Nups at the NE would almost certainly create leaky NPCs (*Popken et al., 2015*; *Strawn et al., 2004*; *Timney et al., 2016*), as also predicted by our models of aged NPCs (*Figure 1—figure supplement 3b*). However, based on our transport data, we can exclude a scenario where leaky NPCs are present at the NE of mitotically aged cells (*Figures 4* and *5*). Instead, our data supports that the misassembled NPCs get covered with membrane (*Figure 3*), a structure known to be present in *nup116*, *vps4pom152* and *apq12* mutant strains (*Scarcelli et al., 2007*; *Webster et al., 2014*; *Webster et al., 2016*; *Wente and Blobel, 1993*). This would prevent loss of compartmentalization but cause an overall reduction of permeable NPCs, which is consistent with the observed decrease in transport dynamics across the NE and the increased steady state compartmentalization (*Figures 4* and *5*). Altogether we propose that consequential to declining quality control of NPC assembly, aged nuclei have fewer functional NPCs and significant amounts of dysfunctional NPCs that do not contribute to the overall transport kinetics (*Figure 6c*).

In a previous study, we generated directional networks to predict how the different changes observed in aging may be interdependent, and proposed that an imbalance affecting the protein biogenesis machinery is the major driving force in yeast replicative aging (*Janssens et al., 2015*). The defects in NPCs in aging may thus be a consequence of the imbalanced protein levels of the NPC components and assembly factors. This imbalance is not transcriptionally driven since, with few exceptions, the transcriptional changes in Nups, NTRs and assembly factors are small (*Janssens et al., 2015*), but is rather related to protein synthesis, folding and assembly. Indeed, the proteome changes in the Nups are well linked to other changes in the proteome of aging cells: Nup116, Nsp1, Nup159, Nic96, Nup82, Nup157 and Nup2 appear in the middle clusters of the proteome network meaning that they are predicted to be consequential to earlier changes, but also that they drive later changes (*Janssens et al., 2015*). Our finding of correlations between remaining lifespan of a cell and the abundance of several Nups and assembly factors, as well as the transport dynamics of Msn2, also supports that a potential causal relation between NPC function and lifespan may exist. The ways that faulty NPCs may further drive aging are numerous. For instance, mitotically aging cells are expected to respond more slowly to changes in their environment due to reduced nuclear cytoplasmic transport and communication. Moreover, declining NPC function, which by itself is a consequence of failing protein quality control, will result in aberrant transcription regulation and

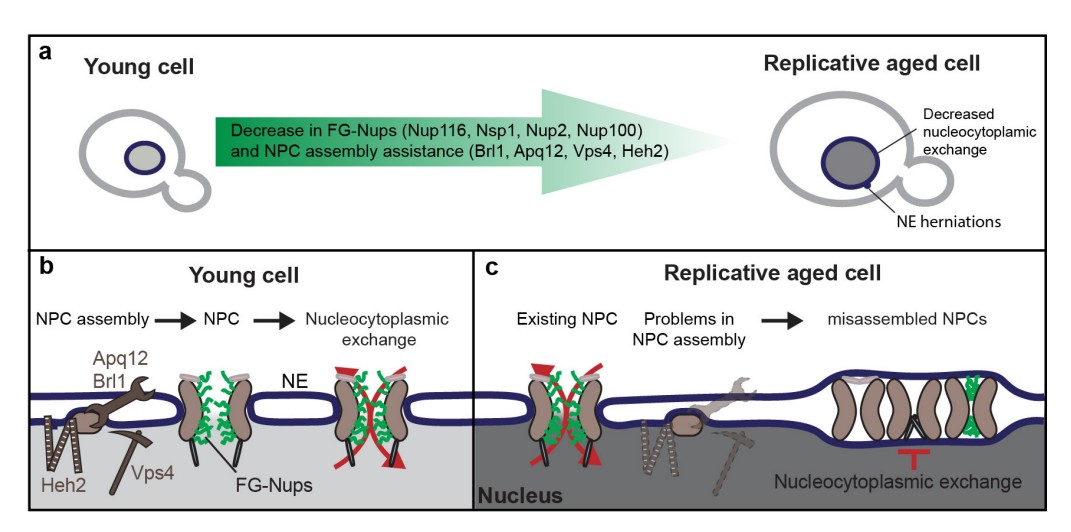

**Figure 6.** Graphical summary. (**a**) Summary of the measured changes. (**b**) Schematic representation of NPC assembly and nuclear transport dynamics in young cells. (**c**) Model: In old cells, the decrease in abundance of several proteins that assist in NPC assembly, and possibly the NPC components themselves, causes the accumulation of misassembled NPCs in aged mother cells. Misassembled NPCs are covered with membrane and do not participate in nucleocytoplasmic exchange, reducing the effective density of transport competent NPCs over time and leading to an increase in steady-state compartmentalization and concomitant decrease in transport dynamics.
DOI: https://doi.org/10.7554/eLife.48186.017

mRNA export, and, with that, contributes to the further loss of protein homeostasis. Lastly, faulty NPCs may impact the loss of genome stability in aging and the joint dependence of NPC assembly and genome stability on Apq12 provides an interesting observation in this context. Altogether, our study contributes to the emerging view that NPC function is a factor of importance in aging and age-related disease (*Fichtman and Harel, 2014*) impinging on the universal hallmarks of human aging of intercellular communication and loss of protein homeostasis.

Our work also provides the first clear example that the assembly of large protein complexes is a major challenge in aging of dividing cells. Indeed, there is data to support that loss of protein complex stoichiometry is a prominent and conserved phenotype in aging (*Janssens et al., 2015*; *Ori et al., 2015*). We speculate that the assembly and quality control of many other long-lived large protein complexes, such as the proteasome and kinetochores, both becoming highly substoichiometric in aging (*Janssens et al., 2015*; *Ori et al., 2015*), are compromised in mitotically aging cells.

## Materials and methods

### Strains

All *Saccharomyces cerevisiae* strains used in this study are listed Table 4 and were validated by sequencing. Experiments in this study were performed with BY4741 genetic backgrounds, except the deletion of *apq12*, which is instable in the BY4741 background. W303 apq12Δ was created by using the PCR-toolbox (*Janke et al., 2004*). The Nup116-GFPboundary MKY227 from *Mattheyses et al. (2010)* was converted from its W303 background to BY4741 background by crossing and tetrad dissection for in total 10 times with BY4741. All strains used in the aging experiments are plasmid free as plasmids are not maintained in aging cells. Cells were grown at 30°C, shaking at 200 RPM using Synthetic Complete medium supplemented with 2% Glucose or 2% D-raffinose, unless indicated otherwise. If applicable, expression of reporter proteins was induced with 0.5% Galactose. Cells were induced for 4–7 hr prior to the start of an experiment. The proteomics data within this manuscript previously published in *Janssens et al. (2015)* is from YSBN6 grown in nitrogen base without amino acids supplemented with 2% glucose.

## Microscopy

All microscopy, excluding the experiments for *Figure 5*, was performed at 30°C on a Delta Vision Deconvolution Microscope (Applied Precision), using InsightSSITM Solid State Illumination of 488 and 594 nm, an Olympus UPLS Apo 60x or 100x oil objective with 1.4NA and softWoRx software (GE lifesciences). Detection was done with a CoolSNAP HQ2 camera. Microscopy to study Msn2 dynamics was performed on a Nikon Ti-E microscope equipped with a Hamamatsu Orca Flash V2 using a 40X oil immersion objective (1.3NA). Fluorescence excitation was performed using an LED illumination system (Excelitas 110-LED) that is triggered by the camera.

## Replicative aging experiments – microfluidic dissection platforms

The microfluidic devices were used as previously detailed (*Crane et al., 2014*; *Lee et al., 2012*). Bright-field images of the cells were taken every 20 min to follow all divisions of each cell. Fluorescent images with three or four z-slices of 0.5 or 0.7 μm, were taken at the beginning of the experiment and after 15, 30, 45 and 60 hr. One experiment lasts for a maximum of 80 hr. All lifespans and the N/C ratios of yPP008, yPP009 and yPP011 (*Figure 4a,b* and *Figure 4—figure supplement 3*) reflect only cells that stay in the device for a whole lifespan are included into the dataset. All other data presented include all cells that stay in the microfluidic device for at least 15 hr and have at least one image well enough in focus for a ratiometric measurement. Data in *Figure 4a,b* and *Figure 4—figure supplement 3* were obtained using both the microfluidic dissection platform (*Lee et al., 2012*) and the ALCATRAS (*Crane et al., 2014*), data in all other experiments were performed using an ALCATRAS chip.

## Poison assay in the microfluidic device

To measure the passive permeability of NPCs in old cells, the cells were replicatively aged in the microfluidic chip for approximately 21 hr. Subsequently, the medium in the chip was exchanged, as described by *Crane et al. (2014)*, for Synthetic Complete medium supplemented with 10 mM sodium azide and 10 mM 2-deoxy-D-glucose (*Shulga et al., 1996*). Additionally, the medium was supplemented with some Ponceau S stain, which makes the medium fluoresce in the mCherry channel. The addition of sodium azide and 2-deoxyglucose depletes the cell of energy and destroyes the Ran-GTP/GDP gradient thus abolishing active transport of reporter proteins. We measured the net efflux of reporter proteins by imaging the cells every 30 s.

## Data analysis of Nups and N/C ratios

Microscopy data was quantified with open source software Fiji (https://imagej.net/Welcome) (*Schindelin et al., 2012*). Fluorescent intensity measurements were corrected for background fluorescence. To quantify the abundance of proteins at the NE, an outline was made along the NE in the mCherry channel. The outline was used to measure the average fluorescent intensities in the mCherry and GFP channels. To quantify the nuclear localization (N/C ratio), the NE was outlined based on the Nup49-mCherry signal and the average fluorescence intensity at the nucleus was measured. A section in the cytosol devoid of vacuoles (appearing black) was selected for determining the average fluorescence intensity in the cytosol. We note that the average fluorescence of GFP in the cytosol may be underestimated in aged cells as aged cells have many small vacuoles that make it hard to select vacuole-free areas in the cytosol. The extent to which this affects the data can best be judged from the cells expression GFP where the N/C ratio on average increases from 1.2 to 1.25 in 30 hr (*Figure 6d*). All heatmaps and bee swarm/box plots were generated in MATLAB (Mathworks https://nl.mathworks.com/).

## EM methods

### Magnetic purification of old cells for electron microscopy

To evaluate the nuclear envelope ultrastructure of replicatively aged yeast, we cultured BY4741 in 200 mL of YPD to mid log phase. $6 \times 10^8$ cells were collected by centrifugation, washed in PBS and then resuspended in 500 μL of 2xPBS. To biotin-label cells, 7 mg of sulfo-NHS-LC-biotin (Pierce) was dissolved in 500 uL of ice-cold $H_2O$ and added to the cell suspension, which was subsequently incubated at RT for 20 min. The cells were pelleted by centrifugation and excess free biotin removed by washing in PBS. Biotin-labeled cells were used to inoculate 4 L of YPD and grown for ~10–12

generations. Cells were collected by centrifugation and resuspended in ice-cold PBS. The cell suspension was incubated with 250 µL of streptavidin-coated magnetic beads (Qiagen) for 20 min at 4° C. Magnetic beads with bound biotinylated cells were collected on a magnet and were washed five times with PBS. The unbound cells were used as the mixed-population (young) sample. After the final wash in PBS, cells were resuspended in a small volume of YPD.

Yeast from either the magnetic bead sorted sample (aged) or non-binding yeast (young) were concentrated into a thick slurry by gently pelleting (2,000 rpm) and aspirating off excess media. The slurries were high-pressure frozen in a Leica HMP100. Frozen samples were freeze-substituted in a Leica Freeze AFS with 0.1% uranyl acetate in dry acetone and infiltrated with Lowicryl HM20 resin. The polymerized resin block was cut with a diamond blade into ~100 nm thick sections. Sections were collected onto a formvar/carbon coated nickel grids and stained with 2% uranyl acetate and Reynolds lead citrate for improved membrane contrast. Images were acquired on a FEI Tecnai Bio-twin TEM at 80 kV equipped with a Moranda CCD camera using iTEM (Olympus) software.

## Msn2:GFP imaging and quantification

Following introduction of the cells to the microfluidic device, brightfield imaging was begun immediately. The process of introducing cells to the device was found to increase Msn2 activity for the first hour or two following device loading. To ensure that the baseline timecourse in young cells was representative of pulse dynamics and not affected by loading stress, fluorescence imaging was begun after cells had been allowed to acclimate for 3 hr. Brightfield images were acquired at every time-point, with intervals of 5 min when fluorescence images were not acquired. Fluorescence images were acquired for 2 hr, at intervals of 90 s, with three z-slices of 1.5 µm. Following the fluorescence imaging, bright-field images were acquired for 8 hr to ensure that cells could be tracked and the number of daughter divisions could be scored. Cells were segmented and tracked using previously published software (*Bakker et al., 2018*).

Nuclear accumulation of Msn2 was quantified using a measure of skewness. Specifically, the ratio of the brightest 2% of the pixels within the cell relative to the median cell fluorescence. By normalizing to the median cell fluorescence, this is measurement is robust to photobleaching or changes in protein concentration. This measurement has been repeatedly validated and used in previous studies of transcription factor translocation dynamics (*Cai et al., 2008*; *Granados et al., 2017*). For each single cell, peaks were located and quantified using the findpeaks function in Matlab. At each age, the measurements for all pulses within a single cell were averaged to generate a single value for mean Msn2 pulse properties for that cell, at that age. This value was used for correlations with remaining lifespan and distribution of pulse widths at each age (*Figure 5—figure supplement 1*). To determine the mean pulse dynamics at each age, all pulses of all cells alive at the age were centered relative to each pulse peak, and averaged.

## Modeling of aged NPCs

In order to model the aged NPC with the measured stoichiometry of FG-Nups from the protein abundance data (see *Figure 1*), we built 24 different models by taking into account the 8-fold symmetry of the NPC. The model details are shown in *Table 1*. In all 24 models, the two peripheral Nsp1's along with Nup116 were deleted. Nsp1 in the central channel recruits Nup49 and Nup57 to form a Nsp1-Nup49-Nup57 subcomplex. As a result, deletion of one of the central channel Nsp1's is accompanied by removal of the corresponding Nup49 and Nup57. We computed the time-averaged radial mass density distribution (density averaged in the circumferential and axial direction) of the FG-Nups for these 24 models along with the wild type (*Figure 1—figure supplement 3*). The average of the 24 models we refer to as the 'Aged proteome' model.

## Force-field parameters of carbonylated amino acids in the 1-bead-per-amino-acid (1BPA) model (*Ghavami et al., 2014*)

Among all amino-acids, Threonine (T), Lysine (K), Proline (P) and Arginine (R) can undergo carbonylation. The change in hydrophobicity upon carbonylation of these amino-acids was calculated with the help of five hydrophobicity prediction programs (KOWWIN, ClogP, ChemAxon, ALOGPS and miLogP) (*Leo, 1993*; *Meylan and Howard, 1995*; *Tetko et al., 2005*; *Viswanadhan et al., 1989*) (Cheminformatics, 2015, www.molinspiration.com). These software programs use the partition

**Table 1.** Details of the FG-Nup stoichiometry for the 24 constructed models to represent the aged NPC. 0 and 1 represent absence and presence, respectively, of the FG-Nup in 8-fold symmetry.

| | NSP1 | NSP1 | NSP1 | NSP1 | Nup1 | Nup42 | Nup49 | Nup49 | Nup57 | Nup57 | Nup60 | Nup100 | Nup145 | Nup145 | Nup159 | Nup116 |
|---|---|---|---|---|---|---|---|---|---|---|---|---|---|---|---|---|
| Model 1 | 0 | 0 | 1 | 1 | 1 | 1 | 1 | 1 | 1 | 1 | 1 | 1 | 1 | 1 | 1 | 0 |
| Model 2 | 0 | 0 | 1 | 1 | 1 | 1 | 1 | 1 | 1 | 1 | 1 | 0 | 1 | 1 | 1 | 0 |
| Model 3 | 0 | 0 | 1 | 1 | 1 | 1 | 1 | 1 | 1 | 1 | 0 | 1 | 1 | 1 | 1 | 0 |
| Model 4 | 0 | 0 | 1 | 1 | 1 | 1 | 1 | 1 | 1 | 1 | 0 | 0 | 1 | 1 | 1 | 0 |
| Model 5 | 0 | 0 | 1 | 1 | 0 | 1 | 1 | 1 | 1 | 1 | 1 | 1 | 1 | 1 | 1 | 0 |
| Model 6 | 0 | 0 | 1 | 1 | 0 | 1 | 1 | 1 | 1 | 1 | 1 | 0 | 1 | 1 | 1 | 0 |
| Model 7 | 0 | 0 | 1 | 1 | 0 | 1 | 1 | 1 | 1 | 1 | 0 | 1 | 1 | 1 | 1 | 0 |
| Model 8 | 0 | 0 | 1 | 1 | 0 | 1 | 1 | 1 | 1 | 1 | 0 | 0 | 1 | 1 | 1 | 0 |
| Model 9 | 0 | 0 | 1 | 0 | 1 | 1 | 0 | 1 | 0 | 1 | 1 | 1 | 1 | 1 | 1 | 0 |
| Model 10 | 0 | 0 | 1 | 0 | 1 | 1 | 0 | 1 | 0 | 1 | 1 | 0 | 1 | 1 | 1 | 0 |
| Model 11 | 0 | 0 | 1 | 0 | 1 | 1 | 0 | 1 | 0 | 1 | 0 | 1 | 1 | 1 | 1 | 0 |
| Model 12 | 0 | 0 | 1 | 0 | 1 | 1 | 0 | 1 | 0 | 1 | 0 | 0 | 1 | 1 | 1 | 0 |
| Model 13 | 0 | 0 | 1 | 0 | 0 | 1 | 0 | 1 | 0 | 1 | 1 | 1 | 1 | 1 | 1 | 0 |
| Model 14 | 0 | 0 | 1 | 0 | 0 | 1 | 0 | 1 | 0 | 1 | 1 | 0 | 1 | 1 | 1 | 0 |
| Model 15 | 0 | 0 | 1 | 0 | 0 | 1 | 0 | 1 | 0 | 1 | 0 | 1 | 1 | 1 | 1 | 0 |
| Model 16 | 0 | 0 | 1 | 0 | 0 | 1 | 0 | 1 | 0 | 1 | 0 | 0 | 1 | 1 | 1 | 0 |
| Model 17 | 0 | 0 | 0 | 1 | 1 | 1 | 1 | 0 | 1 | 0 | 1 | 1 | 1 | 1 | 1 | 0 |
| Model 18 | 0 | 0 | 0 | 1 | 1 | 1 | 1 | 0 | 1 | 0 | 1 | 0 | 1 | 1 | 1 | 0 |
| Model 19 | 0 | 0 | 0 | 1 | 1 | 1 | 1 | 0 | 1 | 0 | 0 | 1 | 1 | 1 | 1 | 0 |
| Model 20 | 0 | 0 | 0 | 1 | 1 | 1 | 1 | 0 | 1 | 0 | 0 | 0 | 1 | 1 | 1 | 0 |
| Model 21 | 0 | 0 | 0 | 1 | 0 | 1 | 1 | 0 | 1 | 0 | 1 | 1 | 1 | 1 | 1 | 0 |
| Model 22 | 0 | 0 | 0 | 1 | 0 | 1 | 1 | 0 | 1 | 0 | 1 | 0 | 1 | 1 | 1 | 0 |
| Model 23 | 0 | 0 | 0 | 1 | 0 | 1 | 1 | 0 | 1 | 0 | 0 | 1 | 1 | 1 | 1 | 0 |
| Model 24 | 0 | 0 | 0 | 1 | 0 | 1 | 1 | 0 | 1 | 0 | 0 | 0 | 1 | 1 | 1 | 0 |

DOI: https://doi.org/10.7554/eLife.48186.018

coefficient $P$ (ratio of concentrations in a mixture of two immiscible phases at equilibrium, usually water and octanol) as a measure of hydrophobicity. Since the range of the ratio of concentrations is large, the logarithm of the ratio of concentrations is commonly used: $\log P_{\text{octanol/water}} = \log \frac{C_{\text{octanol}}}{C_{\text{water}}}$.

These programs provide different estimates of the value of log $P$ for a given chemical structure. They use experimental log $P$ values for atoms or small groups of atoms as a basis and their algorithms are fine-tuned by training with experimental values of complete molecules. The molecules are cut into fragments or into atoms, and their contribution adds up to the log $P$ value of the entire molecule based on the concept of structure-additivity (*Fujita et al., 1964*).

The hydrophobicity scale in the 1BPA force field (*Ghavami et al., 2014*) is derived from three scales that are based on partition energy measurements. Since the free energy of partition is proportional to log $P$, the strategy was chosen to find log $P$ values for the oxidized amino acids. To obtain a reliable value for each of the chemically modified amino acids, a weighted average scheme is used. Instead of using the predicted hydrophobicity for the entire residue, it is more accurate to use the predicted change in hydrophobicity since the change in molecular structure upon introduction of a functional group due to carbonylation is small. To account for the variation in accuracy of the predictor programs, a weight is assigned to each program based on the deviation of the prediction from our existing force field value for the amino-acids in their native state (*Ghavami et al., 2014*). The assigned weight ($w_{k,i}$) for program $k$ for amino acid $i$ is defined as:

$$w_{k,i} = \frac{\left(1/\varepsilon_{k,i}\right)^2}{\sum_{k=1}^{5}\left(1/\varepsilon_{k,i}\right)^2},$$

**Table 2.** Force field parameters for carbonylated amino acids.

Here, $\varepsilon_{\text{Ghavami}}$ and $\varepsilon_{\text{native}}$ represent the hydrophobicity of amino acids in their native condition according to *Ghavami et al. (2014)* and the weighted average scheme, respectively. $\varepsilon_{\text{carbonylated}}$ denotes the hydrophobicity derived from the weighted average scheme and $q_{\text{native to carbonylated}}$ stands for the charge modification from the native to the carbonylated state.

| AA | $\varepsilon_{\text{Ghavami}}$ | $\varepsilon_{\text{native}}$ | $\varepsilon_{\text{carbonylated}}$ | $q_{\text{native to carbonylated}}$ |
|---|---|---|---|---|
| T | 0.51 | 0.52 | 0.34 | 0 –>0 |
| K | 0.00 | 0.00 | 0.59 | 1 –>0 |
| P | 0.65 | 0.63 | 0.43 | 0 –>0 |
| R | 0.00 | 0.07 | 0.43 | 1 –>0 |

DOI: https://doi.org/10.7554/eLife.48186.019

where $\varepsilon_{k,i} = \varepsilon_{k,i} - \varepsilon_{\text{Ghavami}}$ is the difference between the hydrophobicity for amino acid $i$ in the 1BPA force field (*Ghavami et al., 2014*) and that predicted by program $k$. The results are depicted in *Table 2*, showing that K and R become more hydrophobic, whereas T and P become more hydrophilic, compared to their native state.

Carbonylation has additional effects. For instance, the carbonylated form of K and R, that is aminoadipic semialdehyde (Asa) and glutamic semialdehyde (GSA), respectively, loose their positive charge and become neutral, see *Table 2* (*Petrov and Zagrovic, 2011*). In addition, P has a ring structure which opens up during carbonylation making the polypeptide backbone less stiff. We take this into account in our model via the bonded potential (*Ghavami et al., 2013*). The carbonylated form of P and R are the same (GSA) (*Petrov and Zagrovic, 2011*). Therefore, we assign the same hydrophobicity to them, that is 0.43, which is the average of the predicted hydrophobicity of 0.44 and 0.42, respectively. The relevant changes in the force field for carbonylation are summarized in *Table 2*.

To explore the limited effect of carbonylation on the overall distribution of the disordered phase, we analyzed the changes in net hydrophobicity and charge. While the hydrophobicity for T and P is reduced, the hydrophobicity of K and R increases, resulting in only a 5% increase in net hydrophobicity for a maximally carbonylated NPC (see *Table 3*). Furthermore, carbonylation leads to a negatively charged NPC (−7560e) compared to a weakly positive charged wild-type NPC (+512 e) as all K and R become neutral. To separate the effects of charge and hydrophobicity on the structure, we carried out an additional simulation in which we consider only the change in hydrophobicity caused by carbonylation and leave the charge unaffected (termed 'Carbonylated_HP' in *Figure 2—figure supplement 1c–e*). The results show that the carbonylated_HP NPC is more hydrophobic than the wild type (refer to *Table 3*), resulting in a denser FG-Nup network with the maximum at a larger $r$-value. However, when also the charge modification is accounted for in the 'Carbonylated' case in *Figure 2—figure supplement 1d* the Coulombic repulsion leads to a lowering of the density, illustrating that both the change in hydrophobicity and charge affect the distribution of the disordered phase. These changes are small yet noticeable near the scaffold of the NPC, whereas the density at the center ($r < 5$ nm) is hardly affected.

**Table 3.** Physical properties of the wild type, carbonylated and carbonylated_HP NPCs.

In the carbonylated_HP NPC only the effect of carbonylation on the hydrophobicity is accounted for. For the net hydrophobicity, we added the hydrophobicity values $\varepsilon$ of all the residues inside the NPC.

| Force field | +ve Charged AA | -ve Charged AA | Net charge | Net hydrophobicity |
|---|---|---|---|---|
| Wild type | 8072 | 7560 | +512 | 43373.7 |
| Carbonylated | 0 | 7560 | −7560 | 45549.7 |
| Carbonylated_HP | 8072 | 7560 | +512 | 45549.7 |

DOI: https://doi.org/10.7554/eLife.48186.020

**Table 4.** Strains and plasmids

| Yeast strains | Source |
|---|---|
| BY4741 yeast (*MATa his3Δ1 leu2Δ0 met15Δ0 ura3Δ0*) | Invitrogen |
| BY4742 yeast (*MATα his3Δ1 leu2Δ0 lys2Δ0 ura3Δ0*) | Invitrogen |
| Apq12Δ; Y01433 (*MATa; ura3Δ0; leu2Δ0; his3Δ1; met15Δ0; YIL040w::kanMX4*) | This study |
| W303 Apq12Δ *apq12::hphNTI leu2-3, 112 trp1-1 can1-100 ura3-1 ade2-1 his3-11,15* | This study |
| Nup116-GFPboundary MKY227 (W303, ADE2+) | *Mattheyses et al., 2010* |
| Nup116-GFPboundary BY4741 | This study |
| Apq12-GFP yeast (*MATa his3Δ1 leu2Δ0 met15Δ0 ura3Δ0 Apq12-GFP::His3M × 6*) | ThermoFisher (*Huh et al., 2003*) |
| Nup2-GFP yeast (*MATa his3Δ1 leu2Δ0 met15Δ0 ura3Δ0 Nup2-GFP::His3M × 6*) | ThermoFisher |
| Nup49-GFP yeast (*MATa his3Δ1 leu2Δ0 met15Δ0 ura3Δ0 Nup49-GFP::His3M × 6*) | ThermoFisher |
| Nup100-GFP yeast (*MATa his3Δ1 leu2Δ0 met15Δ0 ura3Δ0 Nup100-GFP::His3M × 6*) | ThermoFisher |
| Nup133-GFP yeast (*MATa his3Δ1 leu2Δ0 met15Δ0 ura3Δ0 Nup133-GFP::His3M × 6*) | ThermoFisher |
| Heh2-GFP yeast (*MATa his3 leu2 met15 ura3Δ0 Heh2-GFP::His3M × 6*) | ThermoFisher |
| Srm1-GFP yeast (*MATa his3Δ1 leu2Δ0 met15Δ0 ura3Δ0 Srm1-GFP::His3M × 6*) | ThermoFisher |
| Kap95-GFP yeast (*MATa his3Δ1 leu2Δ0 met15Δ0 ura3Δ0 Kap95-GFP::His3M × 6*) | ThermoFisher |
| Crm1-GFP yeast (*MATa his3Δ1 leu2Δ0 met15Δ0 ura3Δ0 Crm1-GFP::His3M × 6*) | ThermoFisher |
| Msn2-GFP yeast (*MATa his3 leu2 met15 ura3Δ0 Msn2-GFP::His3M × 6*) | ThermoFisher |
| JTY7; Nup49-mCh (*MATα Nup49-mCh::CaURA3 can1Δ::STE2pr-LEU2 ura3Δ0 lyp1Δ leu2Δ0 his3Δ1 met15Δ0*) | *Tkach et al., 2012* |
| yIS010; Nup2-GFP Nup49-mCh (*MATa his3Δ1 leu2Δ0 met15Δ0 ura3Δ0 Nup2-GFP::His3M × 6 Nup49-mCh::URA*) | This study |
| yIS011; Nup100-GFP Nup49-mCh (*MATa his3Δ1 leu2Δ0 met15Δ0 ura3Δ0 Nup100-GFP::Hi3M × 6 s Nup49-mCh::URA*) | This study |
| yIS012; Nup116-GFP Nup49-mCh (*MATa his3Δ1 leu2Δ0 met15Δ0 ura3Δ0 Nup116-GFPboundary Nup49-mCh::URA*) | This study |
| yIS013; Nup133-GFP Nup49-mCh (*MATa his3Δ1 leu2Δ0 met15Δ0 ura3Δ0 Nup133-GFP::His Nup49-mCh::URA*) | This study |
| yIS014; Nup49-GFP Nup133-mCh (*MATa his3Δ1 leu2Δ0 met15Δ0 ura3Δ0 Nup49-GFP::His3M × 6 Nup133-mCh::URA*) | This study |
| yIS018; Apq12-GFP Nup49-mCh (*MATa his3Δ1 leu2Δ0 met15Δ0 ura3Δ0 Apq12-GFP::His3M × 6 Nup49-mCh::URA*) | This study |
| yIS021; Srm1-GFP Nup49-mCh (*MATa his3Δ1 leu2Δ0 met15Δ0 ura3Δ0 Srm1-GFP::His3M × 6 Nup49-mCh::URA*) | This study |
| yIS022; Kap95-GFP Nup49-mCh (*MATa his3Δ1 leu2Δ0 met15Δ0 ura3Δ0 Kap95-GFP::His3M × 6 Nup49-mCh::URA*) | This study |
| yIS023; Crm1-GFP Nup49-mCh (*MATa his3Δ1 leu2Δ0 met15Δ0 ura3Δ0 Crm1-GFP::His3M × 6 Nup49-mCh::URA*) | This study |
| yIS027; Apq12Δ Nup49-mCh GFP-NLS (*MATa; ura3Δ0; leu2Δ0; his3Δ1; met15Δ0; YIL040wΔ::kanMX4; GFP-tcNLS(pGal1)::His Nup49-mCH::URA*) | This study |

*Table 4 continued on next page*

*Table 4 continued*

| Yeast strains | Source |
|---|---|
| yIS028; Apq12Δ Nup49-mCh GFP-NES (MATa; ura3Δ0; leu2Δ0; his3Δ1; met15Δ0; YIL040wΔ::kanMX4; GFP-NES(pGal1):: His Nup49-mCH::URA) | This study |
| yIS032; Chm7-yeGFP Nup49-mCh (MATa his3Δ1 leu2Δ0 met15Δ0 ura3Δ0 Chm7-yeGFP::His Nup49-mCh::URA) | This study |
| yIS035; Heh2-GFP Nup49-mCh (MATa his3Δ1 leu2Δ0 met15Δ0 ura3Δ0 Heh2-GFP::His3M × 6 Nup49-mCh::URA) | This study |
| yPP008; GFP-tcNLS Nup49-mCh (MATa his3Δ1 leu2Δ0 met15Δ0 ura3Δ0 GFP-tcNLS(pGal1)::His Nup49-mCh::URA) | This study |
| yPP009; GFP Nup49-mCh (MATa his3Δ1 leu2Δ0 met15Δ0 ura3Δ0 GFP(pGal1)::His Nup49-mCh::URA) | This study |
| yPP011; GFP-NES Nup49-mCh (MATa his3Δ1 leu2Δ0 met15Δ0 ura3Δ0 GFP-NES(pGal1)::His Nup49-mCh::URA) | This study |
| yAA001; Nab2NLS-GFP Nup49-mCh (MATα Nup49-mCh::CaURA3 can1Δ::STE2pr-LEU2 ura3Δ0 lyp1Δ leu2Δ0 his3Δ1:: Nab2NLS(pTpi1)::His met15Δ0) | This study |
| yAA002; Pho4NLS-GFP Nup49-mCh (MATα Nup49-mCh::CaURA3 can1Δ::STE2pr-LEU2 ura3Δ0 lyp1Δ leu2Δ0 his3Δ1:: Pho4NLS(pTpi1)::His met15Δ0) | This study |
| Oligonucleotides | |
| Chm7-GFP-S3 fw (GAAAACCACGATAATGAG ATAAGAAAAATCATGATGGAAGAACAACCACG TCGTACGCTGCAGGTCGAC) | This study |
| Chm7-GFP-S2 rv (CATATTTATTTTTTATTT ATACATATATATTTATTTATTAGTCACTC AGTTCGATCGATGAATTCGAGCTCG) | This study |
| Plasmids | |
| Plasmid: pYM44 yeGFP-tag | *Janke et al., 2004* |
| Plasmid: pYM28 EGFP-tag | *Janke et al., 2004* |
| Plasmid: pNZ-h2NLS-L-GFP (ID-GFP) | *Meinema et al., 2011* |
| Plasmid: pBT016 pYX242-NAB2NLS-GFP-PRA | *Timney et al., 2006* |
| Plasmid: pBT018 pYX242-PHO4NLS-GFP-PRA | *Timney et al., 2006* |
| Plasmid: pPP014 mCh-Ura-Cassette | This study |
| Plasmid: pPP042 pRS303-GFP-tcNLS | This study |
| Plasmid: pPP043 pRS303-GFP | This study |
| Plasmid: pPP046 pRS303-GFP-NES | This study |
| Plasmid: pAA8 pRS303-Nab2NLS-GFP | This study |
| Plasmid: pAA9 pRS303-Pho4NLS-GFP | This study |

DOI: https://doi.org/10.7554/eLife.48186.021

## Growth of strains for oxidation assays

100 ml of BY4741 expressing Nsp1-GFP, was grown to an $OD_{600}$ of 0.8 after which the culture was split in two portions of 50 ml; one portion was stressed by ROS by the addition of menadione (1 ml of 8 mg/ml in ethanol) for 1.5 hr, while to the other only 1 ml of ethanol was added. The cells were then harvested, washed with water and stored at - 80℃ until use.

## Purification of ID-GFP

*L.lactis* NZ9000 carrying a plasmid from which the yeast Heh2 ID-linker can be expressed as a GFP fusion (ID-GFP)(*Meinema et al., 2011*) was grown in 1 liter of GM17 medium supplemented with Chloramphenicol (5 μg/ml). When an OD600 of 0.5 was reached protein expression was induced by addition of 1 ml of the supernatant of the nisin producing *L. lactis* NZ9700. After 2 hr, the cells were harvested by centrifugation, washed once with 50 mM KPi pH7.0 and the pellet was resuspended in 5 ml of the same buffer. Drops of the suspension were frozen in liquid nitrogen and the resulting frozen droplets were pulverized in a cryomill, cooled with liquid nitrogen. 1.5 grams of the resulting powder was resuspended in 10 ml 100 mM NaPi pH7 150 mM NaCl, 10% glycerol, 0.1 mM MgCl2 5 μg/ml DnaseI, 18 mg/ml PMSF and 5 mM DTT and homogenized with a polytron (2 times 30 s at max speed). The suspension was cleared from non-lyzed cells by centrifugation at 20,000 rcf at 4 degrees for 20 min. 1 ml of Ni-sepharose slurry was pre-equilibrated in a polyprep column (BioRad) with 10 ml of ddH$_2$O and subsequently 10 ml of 100 mM NaPi pH7 150 mM NaCl, 10% glycerol, 5 mM DTT. The cleared lysate was mixed with the equilibrated Ni-sepharose and incubated at four degrees under mild agitation in the polyprep column. The column was subsequently drained, washed with 10 ml 10 ml 100 mM NaPi pH7 300 mM NaCl, 10% glycerol, 15 mM imidazole, 5 mM DTT and 10 ml 10 ml 100 mM NaPi pH7 300 mM NaCl, 10% glycerol, 50 mM imidazole, 5 mM DTT. The bound ID-GFP was finally eluted from the column with 100 mM NaPi pH7 300 mM NaCl, 10% glycerol, 300 mM imidazole. The buffer was exchanged to PBS with a Zebaspin desalting column (Thermo Fisher) and the protein concentration was determined using the BCA kit (Pierce). The protein was stored overnight at 4°C.

## Immunoprecipitation of Nsp1-GFP

Cell lysates were prepared from the cells described above in 0.5 ml lysis buffer (50 mM Kpi pH7, 250 mM NaCl, 1% Triton X100, 0.5% deoxycholate, 1 mM MgCl2, 5 mM DTT and protease inhibitors) with 0.5 mm beads in a Fastprep machine. After bead-beating, the cells were incubated on ice for 15 min, and subsequently centrifuged at 20,000 rcf for 15 min to clear the lysates of beads and unbroken cells. The cell lysates were diluted with lysis buffer to 1 ml and 10 μl of GFP-nanotrap beads (Chromotek) were added. After 1.5 hr of incubation, the beads were washed six times with wash buffer (50 mM Kpi pH7, 250 mM NaCl, 0.1% SDS, 0.05% Triton X100, 0.025% Deoxycholate, protease inhibitors). Bound proteins were subsequently eluted from the beads by adding 20 μl of 10% SDS to the beads and 10 min of incubation at 95°C. As a negative control a cell lysate from BY4741, not expressing any GFP-tagged nucleoporin, was treated as above. As positive controls BY4741 was spiked with 2 μg of purified ID-GFP, or 2 μg of ID-GFP which was first in vitro oxidized with 1 mM CuSO4 and 4 mM H$_2$O$_2$ for 15 min at RT.

## Western blotting and ELISA

Of the eluate, 4 μl was separated on a 10% SDS-PAGE gel, transferred to PVDF membrane and GFP-tagged proteins were detected with anti-GFP antibodies. Of the eluate, 10 μl from the immunoprecipitations was used for the oxi-ELISA. First, the SDS was removed from the sample using the HIPPR detergent removal kit (Pierce), by diluting the sample to 100 μl with PBS, and following the protocol as provided with the kit. The detergent-free protein was subsequently diluted to 1 ml with PBS and two-fold serial dilutions were prepared in PBS. 96-well Nunc maxisorp plates (Thermo Fischer) were coated with 100 μl of the serial dilutions in duplo, and incubated O/N at 4°C. Protein amounts were determined by detection of GFP-tagged bound protein with anti-GFP antibodies and a standard ELISA protocol. In a separate ELISA plate, the carbonylation state of the proteins was assessed with an Oxi-ELISA using the Oxyblott Protein oxidation detection kit (Millipore), which was essentially performed as described in *Alamdari et al. (2005)*.

## Statistical analysis

Statistical parameters including the definitions and exact values of N, distributions and deviations are reported in the Figures and corresponding Figure legends. Significance of changes were determined with a two tailed Student's t-test and with non-overlapping notches indicating 95% confidence that two samples are different. Unless mentioned otherwise, the experimental data coming from at least two independent cultures and microfluidic chip experiments were analysed together. In

specific cases (*Figure 1—figure supplement 2g* and *Figure 2—figure supplement 2*) the datasets deviated due to differences in filter/camera settings and are presented separately.

## Acknowledgements

We are grateful for the gift of strain MKY227 from Sanford Simon from the Rockefeller University. We thank Sara Mavrova, Jelmer de Jong, Elizabeth Carolina Riquelme Barrientos and Wouter Sipma for their help setting up the oxidation studies. We thank the Nathan Shock core at the University of Washington. We thank Morven Graham and Xinran Liu for their EM expertise. We thank Sabeth Verpoorte, Michael Chang and the Veenhoff and Chang labs for their critical input into this project. We are very grateful for a gift from the Ubbo Emmius Fund to the Veenhoff lab. We acknowledge the use of the Peregrine cluster (University of Groningen) and the Cartesius cluster (SURFsara, funding grant by NWO) for the large scale simulations carried out during this project. This project was funded by the Netherlands Organization for Scientific Research (NWO ECHO grant number 711.013.008 to LMV and PO and NWO OPEN grant number ALWOP.2015.053 to LMV), by the Zernike Institute for Advanced Materials (University of Groningen) and the University Medical Centre Groningen. CPL and DJT are supported by grants from the NIH GM 105672 and T32GM007223.

## Additional information

### Competing interests

Matt Kaeberlein: Reviewing editor, *eLife*. The other authors declare that no competing interests exist.

### Funding

| Funder | Grant reference number | Author |
|---|---|---|
| Nederlandse Organisatie voor Wetenschappelijk Onderzoek | ALWOP.2015.053 | Liesbeth M Veenhoff |
| Nederlandse Organisatie voor Wetenschappelijk Onderzoek | ECHO.711.013.008 | Liesbeth M Veenhoff Patrick R Onck |
| National Institutes of Health | NIH GM 105672 | C Patrick Lusk David J Thaller |
| National Institutes of Health | T32GM007223 | C Patrick Lusk David J Thaller |
| University of Groningen | Ubbo Emmius Fund | Liesbeth M Veenhoff |
| University Medical Center Groningen | Graduate Student Fellowship | Patrick R Onck Liesbeth M Veenhoff |
| Nederlandse Organisatie voor Wetenschappelijk Onderzoek | SURFsara | Patrick R Onck |
| University of Groningen | Graduate Student Fellowship | Patrick R Onck |

The funders had no role in study design, data collection and interpretation, or the decision to submit the work for publication.

### Author contributions

Irina L Rempel, Liesbeth M Veenhoff, Conceptualization, Formal analysis, Supervision, Funding acquisition, Investigation, Visualization, Writing—original draft, Project administration, Writing—review and editing; Matthew M Crane, Conceptualization, Resources, Formal analysis, Investigation, Visualization, Methodology, Writing—original draft, Writing—review and editing; David J Thaller, Daniel PM Jansen, Resources, Formal analysis, Investigation, Visualization, Methodology, Writing—original draft; Ankur Mishra, Formal analysis, Investigation, Visualization, Methodology, Writing—original draft; Georges Janssens, Resources, Investigation, Methodology; Petra Popken, Resources, Methodology; Arman Akşit, Resources; Matt Kaeberlein, Resources, Supervision; Erik van der

Giessen, Supervision; Anton Steen, Conceptualization, Formal analysis, Supervision, Investigation, Methodology, Writing—original draft; Patrick R Onck, Conceptualization, Formal analysis, Supervision, Funding acquisition, Investigation, Methodology, Writing—original draft, Writing—review and editing; C Patrick Lusk, Conceptualization, Formal analysis, Supervision, Funding acquisition, Investigation, Writing—review and editing

### Author ORCIDs
Matthew M Crane (iD) https://orcid.org/0000-0002-6234-0954
David J Thaller (iD) http://orcid.org/0000-0003-3577-5562
Arman Akşit (iD) http://orcid.org/0000-0001-9053-701X
Matt Kaeberlein (iD) http://orcid.org/0000-0002-1311-3421
Erik van der Giessen (iD) http://orcid.org/0000-0002-8369-2254
C Patrick Lusk (iD) http://orcid.org/0000-0003-4703-0533
Liesbeth M Veenhoff (iD) https://orcid.org/0000-0002-0158-4728

### Decision letter and Author response
Decision letter https://doi.org/10.7554/eLife.48186.028
Author response https://doi.org/10.7554/eLife.48186.029

## Additional files

### Supplementary files
• Transparent reporting form
DOI: https://doi.org/10.7554/eLife.48186.022

### Data availability
All data generated or analysed during this study are included in the manuscript and supporting files.

The following previously published datasets were used:

| Author(s) | Year | Dataset title | Dataset URL | Database and Identifier |
|---|---|---|---|---|
| Janssens GE, Meinema AC, González J, Wolters JC, Schmidt A, Guryev V, Bischoff R, Wit EC, Veenhoff LM, Heinemann M | 2015 | Yeast transcriptome profiling in replicative ageing | https://www.ebi.ac.uk/arrayexpress/experiments/E-MTAB-3605/ | ArrayExpress, E-MTAB-3605 |
| Janssens GE, Meinema AC, González J, Wolters JC, Schmidt A, Guryev V, Bischoff R, Wit EC, Veenhoff LM, Heinemann M | 2015 | Aging Yeast | https://www.ebi.ac.uk/pride/archive/projects/PXD001714 | EMBL-EBI PRIDE Archive, PXD001714 |

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
