## [Decision Letter]

[Editors’ note: a previous version of this study was rejected after peer review, but the authors submitted for reconsideration. The first decision letter after peer review is shown below.]

Thank you for submitting your work entitled "Age-dependent deterioration of nuclear pore assembly in mitotic cells decreases transport dynamics" for consideration by *eLife*. Your article has been reviewed by three peer reviewers, and the evaluation has been overseen by a Reviewing Editor and a Senior Editor. The following individuals involved in review of your submission have agreed to reveal their identity: Michael P Rout (Reviewer #2).

Our decision has been reached after consultation between the reviewers. Based on these discussions and the individual reviews below, we regret to inform you that your work cannot be considered for publication in *eLife* at this time.

Whereas your paper addresses an important and interesting topic, all three reviewers felt that your paper requires additional experiments in order to bolster your main conclusions. For example, this includes new result that would provide mechanistic insight into how the age-dependent changes at the NPC impact function and additional data on Nup49 and NPC assembly during aging. Since these revisions are expected to take more than 2 months to complete the editorial decision was to reject the manuscript. However, if you decided to address these concerns *eLife* would re-consider your manuscript (as a new submission and review). However, given the extend of the required work you might also want to consider to submit your manuscript elsewhere.

*Reviewer #1:*

Rempel et al. have applied single cell imaging technique to address age-dependent changes in NPC form and function, building off observations originally described in their 2015 *eLife* paper. The concept that NPC function changes in aged cells presented in this manuscript also fits with previous reports of non-functional NPCs being retained in mother cells and changes in NPC transport function during ageing (e.g. Shcheprova et al., 2008, Makio et al., 2013, Denoth-Lippuner et al., 2014, and Lord et al., 2015).

In this paper, data is presented to further this correlate between ageing and NPC function, focusing in part on Nup116 and Nsp1, leading to a conclusion that aged cells show increased compartmentalization based on GFP-NLS/NES reporters and the localization of endogenous shuttling proteins. However, it remains unclear if and/or how nuclear permeability is being impacted to increase compartmentalization. For example, if the amount or types of cargo being actively shuttled changed in ageing cells, could this result in more efficient shuttling of GFP-NLS/NES cargos and the apparent increase in compartmentalization, as compared to the interpretation that permeability is changing? In the specific case of Msn2, which is seen to change shuttling dynamics, how other changes in aged cells impact these dynamics is unclear, making the interpretation of the data in Figure 4 difficult. For example, does the interaction of Msn2 with DNA change causing increased nuclear residence times? Do other modulators that contribute to Msn2 localization change in aged cells, such as cAMP/PKA that influence the sensitivity and frequency of nuclear localization (PMID 12732613). In addition, it is unclear how Nup116 and Nsp1 fit into this phenotype and ultimately if these changes in NPC function are drivers of ageing or simply the consequence of other cellular processes declining in function. These findings are interesting, but without offering direct mechanistic insights, I would not support publication of the current manuscript in *eLife*.

*Reviewer #2:*

In previous work, a number of the authors comprehensively surveyed changes in the transcriptome and proteome of yeast cells as they replicative age, discovering that an increase in proteins that are part of the protein biogenesis machinery relative to their transcripts seemed causally associated with other characteristic ageing phenotypes, and that one of the downstream consequences of this was a loss of stoichiometry for numerous protein assemblies, including the nuclear pore complex (NPC).

Here, these findings on the NPC are extended by a detailed examination of the alterations in its composition and function that occur during aging. The abundance of a significant number of key nucleoporins decreases, both at a whole cell level and (for specifically tested examples) at the nuclear envelope. Importantly, the authors establish that the impact of aging on nuclear transport is largely through an age-dependent impediment in proper NPC assembly, rather than through (as previously indicated by others) oxidative damage to NPC components. The authors present numerous lines of evidence indicating that the accumulating misassembled NPCs appear to be rendered impermeable, thus reducing the effective density of functional NPCs over time, leading to an increase in steady state compartmentalization and concomitant decrease in transport dynamics. This detailed focus on the age-related functional consequences of stoichiometric loss for one particular macromolecular assembly provides a new and key insight into the consequences of ageing of dividing cells.

However, several points need to be clarified by the authors to help the readers navigate the logic of the manuscript and the technical details of the experimental work.

1) The main conceptual point that is not sufficiently clear throughout the text (including the Abstract), potentially leading general readers to confusion, is the causal relationship between aging and NPC malfunction. In previous publications, the authors showed that an imbalance affecting the protein biogenesis machinery seem to be the major driving force in aging, as beautifully summarized in Figure 6B of Janssens et al., 2015. It would be interesting for the authors to put the NPC and nuclear transport machinery in context by indicating where they would localize in such a schematic based on their previous results: is the NPC ranked in the "causal" or "responsive" areas? This would help the reader interpret the results and properly position the NPC defects within the overall aging process.

2) The authors decided to validate the changing Nup levels detected previously in their whole proteome analysis by quantifying the NE signal from fluorescently tagged Nups in aging mitotic cells. The approach is very reasonable, but several points are not clear, could affect the interpretation of the results and should be clarified in the manuscript:

a) What were the criteria used to select the Nups tagged for the fluorescence quantification experiments? Why those and not others? Why not Nsp1, that is specifically referred to later (subsection “The cellular abundance of specific NPC components changes in replicative ageing”, end of last paragraph)? This should be made clear in the text.

b) Why did the authors picked Nup49 as the reference throughout the fluorescence quantification experiments? I noticed that several Nups, crucially including Nup49, seem to be missing from the original dataset and from the plot in Figure 1C. Is this a labeling issue or Nup49 was not included in such analysis? If so, why is the Nup chosen to act as a normalization reference in most of the experiments? Given that it seems the authors lack the initial information to establish it as a reliable and useful control, would it be informative to plot the raw data without normalizing to Nup49 to get an idea of what is the effect of such normalization?

c) According to Figure 4—figure supplement 3, overexpression (and as such should be noted in the text) of basically any construct, seems to affect the lifespan of yeast cells. However, how does tagging of Nups affect such lifespan? It is well known that double tagging could generate synthetic effects that could potentially affect the strain fitness. I would strongly suggest that the authors perform a similar analysis as the one shown in Figure 4—figure supplement 3 for the GFP and mCherry tagged strains used in this study versus a non-tagged (wild type) strain. This way, even if some effects were detected, they could be accounted for in the final plots.

*Reviewer #3:*

In previous studies, the Veenhoff and Heinemann groups have assessed transcriptome and proteome variations in aging populations of yeast cells as a means to interrogate physiological changes in aging mitotic cells. Using data from this previous study, the authors show, in the early figures of this manuscript, that aging populations of cells appear to exhibit changes in protein levels of specific nucleoporins and nuclear transport factors. They suggest that specific changes in Nup levels and the presumed loss of stoichiometry among Nups reflect problems in NPC assembly. As supporting evidence for this conclusion, they present data showing that levels of Heh2-GFP and Apq12-GFP, two proteins that have been previously linked to NE biogenesis and NPC assembly, are reduced relative to Nup49. The authors document direct correlations between a decreased lifespan of individual cells and various parameters such as decreased levels of Nup116 and Nup100, as well as non nups including Heh2, Apq12 and Vps4. Based on these observations, they have attempted to examine nuclear transport changes in individual cells as they age. They showed that steady-state distributions of both import and export cargoes were altered as cells aged, notably showing increased compartmentalization.

Overall, this is an interesting study that has applied innovative single-cell analysis approaches to interrogate NPCs and nuclear transport in aging cells. In my view, the two most significant observations reported are age-dependent variations in Nup protein levels and changes in steady-state distributions of nuclear transport cargos as cells age. However, while these are interesting observations, the study fails to provide any mechanistic insight into the interrelationship between these observations (e.g. do changes in Nups levels of the magnitude detected actually alter nuclear transport?). In addition, the authors' main hypothesis that changes in nuclear transport during aging are caused by defects in NPCs assembly is not supported by direct evidence of increased NPC assembly defects in aging cells (e.g. an increased frequency of NPC 'sealing' with aging). In addition, they have not tested whether the observed alterations in the levels of specific Nups or proteins linked to NPC assembly (such as Vps4 and Apq12) are sufficient for altering NPC assembly or nuclear transport. Finally, is there a causal relationship between changes in NPCs/transport and aging? The authors suggest their studies have established such a relationship; however, I feel that, while their data have established interesting correlations, functional insight is limited.

Several additional points are listed below.

1) No data are presented on the levels of Nup49 during aging (not seen in Figure 1C and not discussed). As this Nup represents an important standard of comparison for various analyses in the manuscript, the authors need to discuss what the proteome analysis of aging cells revealed about Nup49.

2) As suggested above, there is no direct evidence that aging is accompanied by defects in NPCs assembly nor is there any evidence that there is an increase in herniations or the sealing over of NPCs in older cells (e.g. EM analysis). Moreover, while null mutants of heh2, vps4 or apq12 exhibit NPC herniation phenotypes, there is no evidence that decreases in the levels of Heh2, Apq12, and Vps4 observed in the aging cells would compromise NPC assembly. This concern also applies to changes in Nup levels and their effects.

3) The authors state that the "rates of NTR-facilitated-transport (import and export) and passive permeability (influx and efflux) are altered such that in old cells the kinetics of passive permeability is lowered relative to the kinetics of NTR-facilitated-transport." This conclusion is not justified by the data presented. First, no data on the kinetics (or apparent kinetics) of transport are presented in this manuscript to support this conclusion. Measuring steady-state levels of transport cargoes does not provide information on the kinetics of transport. Moreover, the authors conclude that cargo retention mechanisms are minimal contributors to the steady state distribution of cargo. It is not clear to me how the authors make this conclusion. While the data in Figure 4—figure supplement 1 suggest there is no change in efflux rates during aging, this does not address retention.

4) Please provide an interpretation of results show in Figure 1—figure supplement 2C. The data presented in this figure suggest the tag (either GFP or mCh) placed on the Nup (49 or 133) can alter its relative levels during aging.

5) In Figure 3B, the authors state that they exclude outliers; however, the justification for the assignment of these data points as 'outliers' is not clear.

6) In a number of instances, the authors could more accurately reference their statements. For example, the authors reference Webster et al., 2014 for their statement that 'Misassembled NPCs that are induced by mutations are asymmetrically retained, and accumulated in the mother cell over time'. Prior studies performed by Makio et al., 2013, and Colombi et al., 2013, showed Nup mutants retain NPCs in mother cells. Also, regarding the statement: "there are no strains with mutations in the NPC that have been shown to lead to increased steady state compartmentalization", there are report of mutants in Nups that cause increased import in mitotic yeast cells (Makhnevych et al., 2003).

---

## [Author Response]

[Editors’ note: the author responses to the first round of peer review follow.]

Reviewer #1:[…] It remains unclear if and/or how nuclear permeability is being impacted to increase compartmentalization. For example, if the amount or types of cargo being actively shuttled changed in ageing cells, could this result in more efficient shuttling of GFP-NLS/NES cargos and the apparent increase in compartmentalization, as compared to the interpretation that permeability is changing? In the specific case of Msn2, which is seen to change shuttling dynamics, how other changes in aged cells impact these dynamics is unclear, making the interpretation of the data in Figure 4 difficult. For example, does the interaction of Msn2 with DNA change causing increased nuclear residence times? Do other modulators that contribute to Msn2 localization change in aged cells, such as cAMP/PKA that influence the sensitivity and frequency of nuclear localization (PMID 12732613).

The reviewer is concerned that our interpretation that the increased steady state distributions of GFP-reporter proteins is related to the NPCs of aged cells is not the only possible interpretation. We agree that explicit discussion and testing of the alternatives, which had not escaped our attention, is important and we thank the reviewer for pointing this out. The alternative interpretations that we had considered and that we now address and discus (subsection “Increased steady state nuclear compartmentalization in aging is mimicked in NPC assembly mutants”, fourth paragraph) are (i) increasing levels of NTRs (updated Figure 4—figure supplement 2A), (ii) retention of GFP e.g. as a consequence of aggregation (Figure 4—figure supplement 1, Figure 4A, GFP control), (iii) decreasing levels of native cargo (new Figure 4—figure supplement 2E, Figure 4—figure supplement 2E, F).

In addition, to substantiate that NPCs are indeed the problem in ageing we included new data on two more GFP-NLS reporter proteins, namely with the Nab2NLS (Kap104 import cargo), and a Pho4NLS (Kap121 import cargo) (new Figure 4C). The changes in ageing that we observe are similar to those of GFP-cNLS, namely increased steady state nuclear accumulation in aged cells. We further strengthen the link between assembly and transport by showing that Brl1 levels decrease during aging (new Figure 2F) and that NPC assembly problems can increase nuclear compartmentalization in a vps4Δheh2Δ background (new Figure 4F). Most significantly, we now provide direct evidence that aged cells have more herniations (new Figure 3). Altogether, in the new submission we explicitly discuss or test alternative interpretations and provide important additional data supporting the interpretation that altered transport dynamics are related to the NPCs.

Indeed, we agree that it is possible, that Msn2 dynamics (and RCC1, or any other native protein) are influenced by other factors than active transport (most prominently retention) and this is why reporter studies provide a good complement. We state this clearly in the text (subsection “Increased steady state nuclear compartmentalization in aging is mimicked in NPC assembly mutants”, fifth paragraph and subsection “Alterations of the nuclear envelope permeability during aging affects transcription factor dynamics”). With regards to the specific suggestion of altered interactions with the DNA: if that were the case we would have expected to observe an increase in peak prominence – which we don’t observe.

In addition, it is unclear how Nup116 and Nsp1 fit into this phenotype and ultimately if these changes in NPC function are drivers of ageing or simply the consequence of other cellular processes declining in function.

It was unclear to the reviewer how we connect the decreases in Nup116 and Nsp1 to the transport phenotypes observed. Indeed from previous studies using mutants lacking specific FG-Nups (or FGrepeat regions), as well as from our computational modeling of an aged-NPC mimic (Figure 1—figure supplement 3), it is expected that NPCs are more permeable if they would miss some of the FG-Nups that we show are decreasing in abundance in aged cells. Increased permeability is indeed not what we observe. We connect both phenotypes by showing that aged cells have assembly problems (Figure 2G) and increasing numbers of herniation with transport inert NPCs (new Figure 3) and by proposing that aged cells consequently have fewer functional NPCs.

With respect to causality we have indeed not been clear enough and we improved this in the second paragraph of the Discussion section. We now make clear that our findings support the hypothesis that, rather than initiating the ageing process, changes at the NPC further drive the aging process. This is supported by our previous system wide models predicting causal relationships as Nup116 and Nsp1 (and Nup159, Nic96, Nup82, Nup157, Nup2) appear rather in the middle clusters of the proteome network meaning they are predicted to be consequential to earlier changes but also that they are driving later changes.

These findings are interesting, but without offering direct mechanistic insights, I would not support publication of the current manuscript in eLife.

With the extensive additional data now provided, we are the opinion that our data more than sufficiently supports the mechanistic model proposed which was nicely summarized by reviewer 2, “The authors present numerous lines of evidence indicating that the accumulating misassembled NPCs appear to be rendered impermeable, thus reducing the effective density of functional NPCs over time, leading to an increase in steady state compartmentalization and concomitant decrease in transport dynamics." For clarity we summarize the model in the new Figure 6.

Reviewer #2:[…] Several points need to be clarified by the authors to help the readers navigate the logic of the manuscript and the technical details of the experimental work.1) The main conceptual point that is not sufficiently clear throughout the text (including the Abstract), potentially leading general readers to confusion, is the causal relationship between aging and NPC malfunction. In previous publications, the authors showed that an imbalance affecting the protein biogenesis machinery seem to be the major driving force in aging, as beautifully summarized in Figure 6B of Janssens et al., 2015. It would be interesting for the authors to put the NPC and nuclear transport machinery in context by indicating where they would localize in such a schematic based on their previous results: is the NPC ranked in the "causal" or "responsive" areas? This would help the reader interpret the results and properly position the NPC defects within the overall aging process.

We thank the reviewer for this recommendation. We have indeed looked for the presence of Nups, NTRs and assembly factors in the proteome and transcriptome networks to see if they would be present in the more causal or the more responsive clusters.

Apart from five exceptions, the Nups, NTR and assembly factor mRNA profiles are stable in ageing and hence they are not present in the transcriptome network (flat profiles cannot be predictive of changes). Five transcripts are found in the transcriptome network, namely, Nmd5 in cluster 2, Apq12 in cluster 3, Brl1 in cluster 5, Brr6 in cluster 7, and Mlp1 in cluster 8, where cluster 1 is most causal and cluster 9 is most responsive.

Within the proteome network the Nups, NTRs and assembly factors are better represented (especially considering the network is much smaller than that of the transcriptome) and they mostly appear in the more responsive clusters three to five, where cluster 1 is most causal and cluster 5 is most responsive. Specifically, eight Nups are present in proteome network, namely Ndc1 in cluster 2, Nup159 and Nsp1 in cluster 3, Nic96, Nup82 and Nup157 in cluster 4 and Nup116 and Nup2 in cluster 5. Two NTRS (Nmd5, Sce1) and one assembly factor (VPS4) are present in proteome network in clusters 3 and 5.

Altogether, the transcriptional changes in Nups, NTRs and assembly factors are small and thus unlikely important in ageing. The proteome changes in the Nups however are linked to other changes in ageing and are predicted to appear as a response to earlier more causal changes in the proteome, while also driving further changes in ageing. In the Discussion of the current manuscript we make clear that changes at the NPC are predicted to be consequential to earlier changes but also that they drive later changes (Discussion, second paragraph). We also adjusted the Abstract.

2) The authors decided to validate the changing Nup levels detected previously in their whole proteome analysis by quantifying the NE signal from fluorescently tagged Nups in aging mitotic cells. The approach is very reasonable, but several points are not clear, could affect the interpretation of the results and should be clarified in the manuscript:a) What were the criteria used to select the Nups tagged for the fluorescence quantification experiments? Why those and not others? Why not Nsp1, that is specifically referred to later (subsection “The cellular abundance of specific NPC components changes in replicative ageing”, end of last paragraph)? This should be made clear in the text.

We now explain why we picked those Nups specifically and why Nsp1 is not included in the analysis (subsection “The cellular abundance of specific NPC components changes in replicative aging”, second paragraph). Namely, we included Nup116 and Nup2 in our experiments as those Nups showed the strongest decrease in abundance. Nup133 was included because its abundance was stable in ageing and Nup100 was included because it is important for the permeability barrier. We used Nup49-mCh as a reference in all of our microfluidic experiments as Nup49 had previously been used as a marker for NPCs. The proteome data indicated that Nup49 showed a relatively stable abundance profile in aging (new Figure 1—figure supplement 1D). The tagging of the Nups with GFP and mCherry (mCh) reduced the fitness of those strains to different extents but all retained median division time under 2.5 hours (new Figure 1—figure supplement 2B). Nsp1 could not be included in the validation, because the Nsp1-GFP fusion had a growth defect and could not be combined with Nup49-mCh, Nup100-mCh or Nup133-mCh in the BY4741 background.

b) Why did the authors picked Nup49 as the reference throughout the fluorescence quantification experiments? I noticed that several Nups, crucially including Nup49, seem to be missing from the original dataset and from the plot in Figure 1C. Is this a labeling issue or Nup49 was not included in such analysis? If so, why is the Nup chosen to act as a normalization reference in most of the experiments? Given that it seems the authors lack the initial information to establish it as a reliable and useful control, would it be informative to plot the raw data without normalizing to Nup49 to get an idea of what is the effect of such normalization?

We now show the proteome data for Nup49 and three other NPC components that were lacking from Figure 1C (new Figure 1—figure supplement 1D). They were lacking from Figure 1C because they did not make it through the full data processing pipeline used in Janssens et al. Specifically for Nup49 the problem was that the correction that we applied to account for losses of proteins on the bead surface yielded negative values, which precluded further data processing. The raw profiles for both Nup49 replicates are however still useful and shows that Nup49 levels are rather stable in the ageing proteome.

Nup49-mcherry is used as an internal reference. The dilemmas with long time lapse experiments is phototoxicity and hence we chose to only image few z-stacks. Also instrument time is an issue and hence we measured on different setups. As a consequence the absolute intensities are variable due to instrument differences and as the NE is not equally well in focus in all images. Quantifying the ratio’s between two Nups solves these issues and provides a more reliable readout. Relating everything to Nup49-mCherry also solves a third complication, which is that the fluorophore tags GFP and mCherry behave differently in ageing. This was shown in Figure 1—figure supplement 2E where the same Nup tagged with mCherry of GFP show differential absolute increases. In a separate manuscript in preparation we will detail these age-related differences in fluorophore behavior related to pH and maturation. We took out two of the panels from Figure 1—figure supplement 2 because they seem to have caused confusion.

c) According to Figure 4—figure supplement 3, overexpression (and as such should be noted in the text) of basically any construct, seems to affect the lifespan of yeast cells. However, how does tagging of Nups affect such lifespan? It is well known that double tagging could generate synthetic effects that could potentially affect the strain fitness. I would strongly suggest that the authors perform a similar analysis as the one shown in Figure 4—figure supplement 3 for the GFP and mCherry tagged strains used in this study versus a non-tagged (wild type) strain. This way, even if some effects were detected, they could be accounted for in the final plots.

We have indeed found that tagging of the Nups affects the fitness of our strains. Performing proper lifespan on all mutants would represent a significant effort and much microscopy time, which we do not have available. Instead, we have included a panel in Figure 1—figure supplement 2A that compares the median number of divisions during the first 15 h in the chip to give a measure of the fitness of the cells.

Reviewer #3:[…] Overall, this is an interesting study that has applied innovative single-cell analysis approaches to interrogate NPCs and nuclear transport in aging cells. In my view, the two most significant observations reported are age-dependent variations in Nup protein levels and changes in steady-state distributions of nuclear transport cargos as cells age.However, while these are interesting observations, the study fails to provide any mechanistic insight into the interrelationship between these observations (e.g. do changes in Nups levels of the magnitude detected actually alter nuclear transport?). In addition, the authors' main hypothesis that changes in nuclear transport during aging are caused by defects in NPCs assembly is not supported by direct evidence of increased NPC assembly defects in aging cells (e.g. an increased frequency of NPC 'sealing' with aging). In addition, they have not tested whether the observed alterations in the levels of specific Nups or proteins linked to NPC assembly (such as Vps4 and Apq12) are sufficient for altering NPC assembly or nuclear transport.

In the new manuscript we better clarify and support that NPC assembly defects can connect the observations of the age-dependent variations in Nup protein levels and the changes in steady-state distributions of nuclear transport cargos as cells age. In the new version of the manuscript, we additionally show that Brl1 levels decrease during aging (new Figure 2F) and that NPC assembly problems can increase nuclear compartmentalization in a vps4Δheh2Δ background (new Figure 4F). Most significantly, we now provide direct evidence that aged cells have more NE herniations (new Figure 3).

The reviewer is further concerned that the observed alterations in the levels of specific Nups or proteins linked to NPC assembly may not be sufficient to alter NPC assembly or nuclear transport. Addressing this experimentally would require creating a strain where the expression levels of multiple Nup and four assembly factors is altered simultaneously (an aged-NPC mimic in young cells); this clearly falls beyond the scope of this project and may in fact be close to impossible to do.

Finally, is there a causal relationship between changes in NPCs/transport and aging? The authors suggest their studies have established such a relationship; however, I feel that, while their data have established interesting correlations, functional insight is limited.

This is also discussed in our answers to reviewer 2 point 1: we have clarified in the Discussion that the current data and the previous system wide study suggest that NPC defects rather further drive the aging process than initiating it.

Several additional points are listed below.1) No data are presented on the levels of Nup49 during aging (not seen in Figure 1C and not discussed). As this Nup represents an important standard of comparison for various analyses in the manuscript, the authors need to discuss what the proteome analysis of aging cells revealed about Nup49.

We have included this data in Figure 1—figure supplement 1D; please see also our explanation to reviewer 2.

2) As suggested above, there is no direct evidence that aging is accompanied by defects in NPCs assembly nor is there any evidence that there is an increase in herniations or the sealing over of NPCs in older cells (e.g. EM analysis). Moreover, while null mutants of heh2, vps4 or apq12 exhibit NPC herniation phenotypes, there is no evidence that decreases in the levels of Heh2, Apq12, and Vps4 observed in the aging cells would compromise NPC assembly. This concern also applies to changes in Nup levels and their effects.

This important point is addressed in response to reviewer 3’s initial comments.

3) The authors state that the "rates of NTR-facilitated-transport (import and export) and passive permeability (influx and efflux) are altered such that in old cells the kinetics of passive permeability is lowered relative to the kinetics of NTR-facilitated-transport." This conclusion is not justified by the data presented. First, no data on the kinetics (or apparent kinetics) of transport are presented in this manuscript to support this conclusion. Measuring steady-state levels of transport cargoes does not provide information on the kinetics of transport. Moreover, the authors conclude that cargo retention mechanisms are minimal contributors to the steady state distribution of cargo. It is not clear to me how the authors make this conclusion. While the data in Figure 4—figure supplement 1 suggest there is no change in efflux rates during aging, this does not address retention.

We have adjusted the mentioned statement and explicitly discuss the meaning of the steady state measurements in the third paragraph of the subsection “Increased steady state nuclear compartmentalization in aging is mimicked in NPC assembly mutants”. Also we added additional data on two more GFP-NLS reporters (new Figure 4D). What is requested, direct kinetic measurements of import and export *in ageing* is technically very challenging and impossible at present. The efflux experiments show to the least that the GFP reporters do not aggregate in the nucleus of aged cells (and that the NE is not more permeable).

4) Please provide an interpretation of results show in Figure 1—figure supplement 2C. The data presented in this figure suggest the tag (either GFP or mCh) placed on the Nup (49 or 133) can alter its relative levels during aging.

We have removed this panel as it caused confusion, also for reviewer 2. We have no reason to think that the tag alters the actual protein levels in ageing. We do have data to show that the two fluorophores themselves have different behaviors in ageing due to differences in pH and maturation time. As mentioned before, we are preparing a manuscript to detail these effects.

5) In Figure 3B, the authors state that they exclude outliers; however, the justification for the assignment of these data points as 'outliers' is not clear.

We have clarified in the manuscript that no outliers were excluded from the analysis. We merely plotted points which were outliers individually. Outliers, are data points outside of the 75^th^/25^th^ percentile +/-

1.5 times the inter quartile range.

6) In a number of instances, the authors could more accurately reference their statements. For example, the authors reference Webster et al., 2014 for their statement that 'Misassembled NPCs that are induced by mutations are asymmetrically retained, and accumulated in the mother cell over time'. Prior studies performed by Makio et al., 2013, and Colombi et al., 2013, showed Nup mutants retain NPCs in mother cells. Also, regarding the statement: "there are no strains with mutations in the NPC that have been shown to lead to increased steady state compartmentalization", there are report of mutants in Nups that cause increased import in mitotic yeast cells (Makhnevych et al., 2003).

We apologize for these mistakes and have corrected our statements and included the reference in our manuscript.